

**Effects of topographic and meteorological parameters**

**on the surface area loss of ice aprons in the Mont-Blanc massif (European Alps)**

Suvrat Kaushik [1,2], Ludovic Ravanel[1,3], Florence Magnin[1], Yajing Yan[2], Emmanuel Trouve[2], Diego Cusicanqui[4]

[1] EDYTEM, Univ. Savoie Mont-Blanc, Univ. Grenoble Alpes, CNRS, 73000 Chambéry, France

[2] LISTIC, Univ. Savoie Mont Blanc, Polytech, F-74944 Annecy-le-Vieux, France

[3] Department of Geosciences, University of Oslo, Sem Sælands vei 1, 0371 Oslo, Norway

[4] IGE, Univ. Grenoble Alpes - CNRS, F-38000 Grenoble, France

**Correspondence:** Suvrat Kaushik (suvrat.kaushik@univ-smb.fr)

**Abstract**

Ice aprons (IAs) are part of the critical components of the Alpine cryosphere. As a result of the changing climate over the past few decades, deglaciation has resulted in a surface decrease of IAs, which has not yet been documented out of a few specific examples. In this study, we quantify the effects of climate change on IAs since the mid-20[th] century in the Mont-Blanc massif (western European Alps). We then evaluate the role of climate forcing parameters and the local topography in the behaviour of IAs. We precisely mapped the surface areas of 200 IAs using high-resolution aerial and satellite photographs from 1952, 2001, 2012 and 2019. From the latter inventory, the surface area of the present individual IAs ranges from 0.001 to 0.04 km$^2$. IAs have lost their surface area over the past 70 years, with an alarming increase since the early 2000s. The total area, from 7.93 km$^2$ in 1952, was reduced to 5.91 km$^2$ in 2001 (-25.5 %) before collapsing to 4.21 km$^2$ in 2019 (-47 % since 1952). We performed a regression analysis using temperature and precipitation proxies to understand better the effects of climate forcing parameters on IA surface area variations. We found a strong correlation between both proxies and the relative area loss of IAs, indicating the significant influence of the changing climate on the evolution of IAs. We also evaluated the





role of the local topographic factors in the IAs area loss. At a regional scale, factors like direct solar radiation and elevation have an important influence on the behaviour of IAs, while others like curvature, slope, and size of the IAs seem to be rather important on a local scale.

**Key words:** Ice aprons, surface area loss, topographic factors, meteorological parameters, Mont Blanc massif.

**1 Introduction**

The predicted shift in climate dynamics over the next decades will undoubtedly have severe consequences on the high mountain environments, primarily on glacier extent (Rafiq and Mishra, 2016; Kraaijenbrink et al., 2017; IPCC, 2021), permafrost (Magnin et al., 2017) and ice and snow cover (Rastner et al., 2019; Guillet and Ravanel, 2020). The effects of climate change on glaciers constitute a remarkably well-discussed topic in the scientific community (Yalcin, 2019).

Climatic factors (mainly temperature and precipitation) are the main driving forces responsible for these changes (Scherler et al., 2011; Bolch et al., 2012; Davies et al., 2012). Shifting temperature and precipitation trends leads to the advance or retreat of glaciers both in volume and surface area (Liu et al., 2013; Yang et al., 2019). On a regional and global scale, many authors have studied the impacts of climate warming on glacier retreats and, consequently, on the hydrology of the mountain environments (*e.g.*, Baraer et al. (2012); Sorg et al. (2014); Frans et al. (2016); Coppola et al. (2018).

However, as observed by Furbish and Andrews (1984), Oerlemans et al. (1998), Hoelzle et al. (2003) and Salerno et al. (2017), glaciers present in the same climate regime can respond to climate change in different ways. The local climate variations can partly explain these variable responses. However, many of these variations result from different morphometric (size, shape, length) and topographic (altitude, slope, aspect, curvature, terrain ruggedness) characteristics.

Several studies have been devoted to understanding the linkage between topographic factors and the response of glacier/ice bodies (*e.g.,* Davies et al., 2012; De Angelis, 2014; Salerno et al., 2017).





World Glacier Monitoring Service (WGMS) monitors glacier changes in all the major mountain regions of the world. However, most mapping and monitoring studies on a global scale focus on massive glaciers since they are generally assessable and easier to monitor compared to other ice features (Liu et al., 2013).

Studies are rare for small glaciers or ice bodies, which generally show a more pronounced response to climate change (Oerlemans and Reichert, 2000; Triglav-Čekada and Gabrovec, 2013; Fischer et al., 2015). This has led to a critical gap in our understanding of their behaviour and mass balance estimates. As part of this trend, ice aprons (IAs), sometimes also referred to as 'rock faces partially covered with ice' (Gruber and Haeberli, 2007; Hasler et al., 2011), have also received poor attention from the scientific community.

These small ice accumulations on steep rock slopes are commonly found in all significant glacierized basins worldwide. However, a concrete and well-summarised definition for IAs is still missing from the literature. Previously, many authors like Benn and Evans (2010), Singh et al. (2011) and Cogley et al. (2011) tried to define IAs, but the most precise definition for IAs up to now can be found in Guillet and Ravanel (2020) for the Mont-Blanc massif (MBM; European Alps). These authors defined IAs as "very small (typically smaller than 0.1 km$^2$ in extent) ice bodies of irregular outline, lying on slopes >40º, regardless of whether they are thick enough to deform under their weight". The small spatial extent of the IAs makes them very difficult to map and monitor. Also, they are typically present in extremely challenging topographies on isolated steep slopes. Cogley et al. (2011) specified that IAs are "lying above the head of a glacial bergschrund which separates the flowing glacier ice from the stagnant ice, or a rock headwall".

Because of their presence on steep slopes, IAs are essential natural elements for the practice of mountaineering, especially in famous destinations like MBM (Barker, 1982). IAs are passing points for most classic mountaineering routes (Mourey et al., 2019). Hence, the loss of IAs is a severe threat to the iconic practice of mountaineering, inscribed in 2019 by UNESCO on the Representative List of the Intangible Cultural Heritage of Humanity. IAs on steep rock walls also carry the critical role of covering steep rock slopes and preventing them from direct exposure to direct solar radiation, thus partly preventing the warming of the underlying permafrost. In addition, a recent study by Guillet et al. (2021) showed that the ice present at the base of the Triangle du Tacul IA could be older than 3 ka, making IAs a potential important glacial heritage.





Guillet and Ravanel (2020) showed that IAs in the MBM have lost mass since the Little Ice Age
(LIA). Based on six different IAs, their study also showed an acceleration in the shrinkage since
the 1990s. They linked the loss of IA area with meteorological parameters, mainly air temperature
and precipitation. It was thus the first documented evidence that IAs have been losing ice volume
due to the changing climate. However, since this study was local and based on only a few IAs, the
authors could not consider other factors such as the local topography, critical for small glacier
bodies (Hock, 2003; Laha et al., 2017).
Thus, to overcome these limitations, we propose a global analysis to ascertain the relationship of
the area loss of IAs with the meteorological parameters, mainly air temperature and precipitation,
using a more comprehensive database (*c*. 200 IAs) covering the whole MBM. The large inventory
of IAs has been surveyed thanks to high-resolution aerial and satellite images from 1952, 2001,
2012 and 2019. Further, based on our inventory, we also evaluate the impacts of the
topographic/geometric controls on the area changes of IAs. For this, we consider the size of IA,
elevation/altitude, slope, curvature, Topographic Ruggedness Index (TRI), direct solar radiation
and permafrost conditions (classified together as topographic factors) based on past studies on
similar themes (*e.g.*, Oerlemans et al., 1998; Warren, 2008; DeBeer and Sharp, 2009; Jiskoot et
al., 2009; Davies et al., 2012; Salerno et al., 2017).


**2 Study area and the impacts of climate change in the region**

The Mont-Blanc massif (Fig. 1) is located in the north-western (external) Alps between France,
Switzerland, and Italy. It covers *c.*550 km$^2$ and displays some of the highest peaks in the European
Alps; a dozen peaks have elevations greater than 4000 m a.s.l. MBM thus shows a significant
variation in the elevation range throughout the massif; the lowest point of the massif is at 1050 m
a.s.l. (Chamonix) and the highest, the top of Mont Blanc at 4808 m a.s.l.
Because of its high elevation, the MBM is also the most glacierized massif in the French Alps
(Gardent et al., 2014). There are about 100 glaciers often bordered by steep rock walls, including
12 glaciers larger than 5 km$^2$. The steep and irregular terrain facilitates the development of many
unique ice bodies like cold-based hanging glaciers or IAs. As a result of an asymmetry of the
massif, 6 of its largest glaciers are located on its NW French side, where slopes are gentler than



the Italian side and glaciers are well fed by the westerly winds while melting is reduced by the
protection of the shaded North faces. The SE Italian side is characterized by smaller glaciers and
generally steeper slopes bounded by very high sub-vertical rock walls. This asymmetry is also
evidenced by the difference in climatic conditions observed on the two sides of the massif. For
example, the Mean Annual Air Temperature (MAAT) recorded in Chamonix (at 1044 m a.s.l.) is
+7.2ºC while that in Courmayeur (1223 m a.s.l.) is + 10.4ºC (Deline et al., 2012). Comparing the
annual MAAT values from 1934 to today shows that MAAT increased by > 2.1ºC in Chamonix
(*MétéoFrance* data). Moreover, the increase in MAAT from 1970 to 2009 was almost four times
faster than from 1934 to 1970 (Mourey et al., 2019). Not only at lower elevations, but the MAAT
also increased by 1.4ºC at elevations exceeding 4000 m a.s.l. between 1990 and 2014 (Gilbert and
Vincent, 2013). The MBM has experienced nine summers characterized by heatwaves (where
maximum temperatures for at least three consecutive days exceed a heatwave temperature
threshold defined for the region) since 1990 (1994, 2003, 2006, 2009, 2015, 2017, 2018, 2019 and
2020), with the one as recent as 2018 being the second (after 2003) hottest. The average annual
precipitation recorded for Chamonix is 1,288 mm, and 854 mm for Courmayeur (Vincent, 2002).
The precipitation rates in the MBM have remained relatively constant since the end of the LIA,
but there is a noticeable decrease in the number of snowfall days relative to the total precipitation
days below 2700 m a.s.l. (Serquet et al., 2011).
Global warming has led to a general retreat trend of the MBM glaciers since the end of the LIA
despite small re-advances culminating in 1890, the 1920s and the 1980s (Bauder et al., 2007). The
recorded loss of glacier surface area was 24 % of the total area from the end of the LIA to 2008
(Gardent et al., 2014). The reported loss of ice thickness is also noteworthy. For example, the loss
of ice thickness at the front of the "Mer de Glace" glacier (1650 m a.s.l.) from 1986 to 2021 is 145
m; the Argentière glacier (1900 m a.s.l.) has lost 80 m in thickness from 1994 to 2013 (Bauder et
al., 2007). At 3550 m a.s.l., the surface of the Géant glacier also lowered by 20 m between 1992
and 2012 (Ravanel et al., 2013). The glacier retreat and shrinkage concur with the Equilibrium
Line Altitude (ELA) that rose by 170 m between 1984 and 2010 in the western Alps (Rabatel et
al., 2013). As a result of the loss of ice volume, the density of open crevasses has considerably
increased, along with an increase in bare ice areas. In some instances, loss of ice volume leads to
instability of steep slopes, and serac falls from the front of warm and cold glaciers are more
frequent (Fischer et al., 2006). This latter process can be typical during the warmest periods of the



year (Deline et al., 2012). Warming trends also intensify moraine erosion, leading to an increased
frequency of rockfalls and landslides (Deline et al., 2015; Ravanel et al., 2018).
Degradation/warming is another critical concern for permafrost (*e.g.*, Haeberli and Gruber, 2009).


**3 Data description**

This section describes all the datasets obtained from diverse sources used in this study (Table. 1).

**3.1 Digital Elevation Model**

Since one of the main aims of our study is to perform a joint analysis of the behaviour of small ice
bodies and the local topography, it was paramount to have a robust high resolution and accurate
Digital Elevation Model (DEM) for the study region. To avoid the uncertainties that most global
DEMs are plagued with and overcome the problem of different DEM origins on the French and
Italian sides of the MBM, we built our own DEM. As part of the CNES *Kalideos Alps* project,
stereoscopic sub-meter resolution optical images from the Pleiades constellation were acquired.
Using the pair of stereo panchromatic images (25/08/2019), a 4 m resolution DEM was computed
using the Ames Stereo Pipeline (ASP), an open-source processing chain developed by Shean et al.
(2016). ASP uses the rational polynomial coefficients provided with the Pleiades images,
eliminating the requirement of many high accuracy ground control points (GCPs). Parameters used
for the processing were kept the same as Marti et al. (2016). ASP generates a point cloud with
elevation values, then gridded into a 4 m resolution DEM using the 'point2DEM' function. The
robustness and efficiency of the ASP processing chain for Pleiades data processing based on GCPs
has been documented previously by Berthier et al. (2007, 2014). The second part of the processing
involves accurately co-registering the newly built DEM with an existing reference DEM of high
precision and accuracy. For this purpose, we used the automatic DEM co-registration methodology
given by Nuth and Kääb (2011). As a 'reference', we used a 2 m LiDAR DEM (Fig. 2a) built by
the *Institut des Géosciences de l'Environnement* (IGE) to co-register the 'source' 4 m Pleaides
DEM (Fig. 2b) generated in the previous step. We used the RGI (Randolf Glacier Inventory)
glacier contours, forest extensions and manually delineated polygons to mask non-stable areas like





glacier and forest regions with an open-source script (https://github.com/dshean/demcoreg). After
masking the non-stable areas, only stable areas were used for the co-registration process. The
source DEM was then shifted (translation only) using the corresponding shift values (in meters)
for x, y and z. A precisely co-registered, high-resolution, robust 4 m DEM was obtained at the end
of the processing. We used this DEM to compute topographic parameters like slope, aspect,
curvature, elevation, TRI, MARST and direct solar radiation.


**3.2 Optical aerial and satellite images**

This study relies on high-resolution aerial and satellite images (Table 1). Working with data from
different sources allows us to tap into the wealth of data for comparison. Spanning over seven
decades and covering the whole MBM ortho-images for 1952, 2001 and 2015 at 0.2 m resolution
were downloaded from *Géoportail* IGN (French *Institut national de l'information géographique*
*et forestière*), while the panchromatic and XS images from SPOT 6 and Pleiades at 2.2 m and 0.5
m respectively were downloaded from the *Kalideos Alps* website. Considering the small
dimensions of the ice bodies, we could only work with high-resolution optical images covering
the entire MBM. We were thus limited by only one set of excellent quality images for 1952 and
2001, as very high-resolution images for this study period were not available from any other
source. For 2012 and 2019, we have data from multiple sources to deal with the problems
associated with the lack of coverage, cloud cover, illumination, shadow, and seasonal snow cover
that make visual interpretation difficult. To avoid overestimating the extent of IAs, it was
preferable to have all images acquired at the end of the summer period (late August or early
September). Considering that our optical images come from many sources, it is necessary to
accurately co-register all images. We used the automatic 'image to image' co-registration tool in
ENVI 5.6. The process includes locating and matching several feature points called tie points in a
'reference' image and a 'warped' image selected for co-registration. Here, we used the Pleiades
panchromatic image of 2019 as a reference, and all the warped images were accordingly co-
registered. Both coarse and fine co-registration procedures were performed, and the co-registration
process was stopped when the RMSE values achieved were less than half the pixel resolution of





the warped image based on the recommendations of Han and Oh (2018). A more detailed
description of the co-registration process was discussed in Kaushik et al. (2021).

**3.3 Meteorological data**

To explore the correlated variations in the surface area of IAs with the changing climate, we need
to build proxies to define accumulation and ablation phases. A similar study for 6 IAs was
performed by Guillet and Ravanel (2020); we aim to test the validity of their results with a more
extensive database (*c*.200 IAs) in the entire MBM. Since the IAs are spread across different
elevation ranges, we test the results using the SAFRAN reanalysis product (Vernay et al., 2019)
that produces gridded datasets of temperature, precipitation, wind speed, and other meteorological
variables at an hourly time step. These data are available as NetCDF files from 1958 for all the
French massifs, at every 300 m elevation belts, at 0, 20, 40° slopes, and for all eight aspects (N,
NE, E, SE, S, SW, W, NW).
For our analysis, we used weather records from the Col du Grand Saint Bernard (GSB), located
close to the MBM at 2469 m a.s.l., and from the Aiguille du Midi (AdM) cable-car station (3810
m a.s.l.). GSB represents a similar climatological regime as the MBM, and the weather records are
available for an extended period starting from the 1860s. Such long-term weather records are not
available from any weather station in the MBM. Since all IAs are present at elevations above the
elevation of the GSB weather station, it was necessary to transform the weather records to an
elevation closer to the average elevation range of the IAs. For this reason, it was necessary to
transform the data from the GSB station using the weather records from the AdM weather station
(data available since 2007). Guillet and Ravanel (2020) found a strong correlation between the
monthly averaged AdM and GSB temperature records and were able to transform the GSB
temperatures using a linear model:

$$T_{AdM} = \alpha T_{GSBi} + \beta + r_i, \qquad (1)$$

where $\alpha = 0.87$ (slope) , $\beta = -7.7^o$ C (intercept) and r (residuals) with zero mean.
No transformation for the precipitation values was performed as this relation is tough to establish
and not always linear (Smith, 2008). Hence, the original GSB precipitation values were used for





the analysis. Using these weather records, Guillet and Ravanel (2020) found a  robust correlation
between ablation and accumulation proxies and the surface area change of 6 IAs. We used the
same datasets to test for similar potential relationships for *c.* 200 IAs, and the results are shown in
Sect. 5.5.
Since the previous study involved a small number of IAs, the disparity arising from elevation
differences of IAs (in turn, the temperature and precipitation coming from weather stations at a
fixed elevation) could have been minimized or not well represented. We decided to use the
SAFRAN reanalysis product and checked for similar potential relationships of climate variables
with the surface area change of IAs. The first problem we encountered was that the SAFRAN data
starts from 1958, while our first images date from 1952. Therefore, for comparison, it was essential
to interpolate the missing data for the six years before 1958 (Fig. 3).  Like the previous
methodology, we looked for a linear relationship between the SAFRAN temperature data (at 2400
m a.s.l. elevation belt) and the GSB temperature data. We again found a strong correlation between
the two datasets (Fig. 4) which helped us transform the data using:

$$T_{SAFRAN2400} = \alpha \, T_{GSB \, i} + \beta + r_i, \quad\quad\quad (2)$$

where $\alpha = 1.01$ (slope) , $\beta = -1.35^{\circ}$ C (intercept) and r (residuals) with zero mean.
For the SAFRAN data estimated (2400 m a.s.l.) from 1952, it was essential to extrapolate the data
for all elevation bands. We used a standard gradient of  $-0.53^{\circ}$ C/100 m increase of elevation based
on the observations of Magnin et al. (2015) for the MBM.
As previously stated, a similar relationship for precipitation is tough to establish. Hence, for the
analysis, we used the SAFRAN precipitation data from 1958 and extrapolated the precipitation
values from the GSB weather station to all elevation bands of SAFRAN data before 1958 (six
years, up to 1952). However, taking a cue from the previous study of Guillet and Ravanel (2020),
we expect the impact of this to be insignificant when considering the results over seven decades.


**4 Methods**

**4.1 Mapping the surface area of IAs from high-resolution satellite images**




IAs boundaries were manually delineated/digitized by the first author of this paper to maintain

data consistency in a GIS environment for 1952, 2001, 2012 2019. The problem of seasonal snow,

which can lead to an overestimation of surface areas, was avoided by using images at the end of

the ablation period. The differentiation of IAs from other snow/ice bodies relies on the slope angle

(we only consider ice bodies on slopes $> 40^\circ$ to be IAs) and whether they are thick enough to

deform under their own weight and show movement like in the case of hanging glaciers. The slope

mask to remove areas with slopes $< 40^\circ$ was built in ArcGIS 10.6 using the Pleaides DEM.

Figure 5 shows the variations in the surface areas of IAs over the study period. It also highlights

the importance of high-resolution images because of the small dimensions of the ice bodies we are

studying.

**4.2 Generation of topo-climatic parameters**

The relative area loss of IAs for three time periods, *i.e.* 1952 to 2001, 2001 to 2012 and 2012 to

2019, is analyzed with all topographic factors. The area loss is expressed as a relative percentage

of the area lost between the first observation and the next. Authors like Salerno et al. (2017) have

also used absolute values, but for our study, this would not give a fair estimation for the analysis

as it generates a bias based on the size of IAs. The factors we considered for our analysis are

elevation, slope, aspect, curvature, TRI, direct solar radiation (all estimated in ArcGIS 10.6), mean

annual rock surface temperature (MARST), and size of the IAs. The topographic parameters are

generated using the 4 m Pleaides DEM (see Sect. 3.1.1).

**Direct solar radiation:**

Direct solar radiation (DSR) measures the potential total insolation across a landscape or at a

specific location. The intensity of solar radiation received at the surface mainly depends on the

latitude and time of the year. On a local scale, components such as topographic shading, slope, and

aspect also control the radiation distribution (Olson and Rupper, 2019). The viewshed algorithm

was run based on a uniform sky and a fixed atmospheric transmissivity value of 1. Sabo et al.

(2016) showed the application of these algorithms in areas of rough topography. The total DSR



(DSR$_{tot}$) for a given location is calculated as the sum of the DSR (Dir$_{\theta,\alpha}$) from all the sun sectors
(calculated for every sun position at 30 minutes intervals throughout the day and month for a year):
$$DSR_{tot} = \sum DSR_{\theta,\alpha} \qquad (3)$$


The direct solar radiation (Dir$_{\theta,\alpha}$) with a centroid at zenith angle ($\theta$) and azimuth angle ($\alpha$) is
calculated using the following equation:

$$DSR_{\theta,\alpha} = S_{Const} * (\beta^{m(\theta)}) * SunDur_{\theta,\alpha} * SunGap_{\theta,\alpha} * cos(AngIn_{\theta,\alpha}), \qquad (4)$$


where $S_{Const}$ is the solar constant with a value of 1367 W/m$^2$, $\beta$ is the transmissivity of the
atmosphere (averaged over all wavelengths) for the shortest path (in the direction of the zenith),
$m(\theta)$ is the relative optical path length, measured as a proportion relative to the zenith path length,
SunDur$_{\theta,\alpha}$ is the time duration represented by the sky sector, SunGap$_{\theta,\alpha}$ is the gap fraction for the
sun map sector and AngIn$_{\theta,\alpha}$ is the angle of incidence between the centroid of the sky sector and
the axis normal to the surface.
The final map of DSR is the sum of values calculated at an hourly time step for every pixel, as per
the resolution of the DEM used. The values of solar radiation are given in W/m$^2$. Higher values
for solar radiation indicate higher insolation, while lower values suggest low insolation. We prefer
DSR over the aspect for our analysis to avoid bias due to local shading on sun-exposed faces,
considering the slope angle associated with the aspect.

**Elevation:**
Elevation strongly influences the climatic conditions within the same region, significantly altering
the precipitation, temperature, and wind regime even at a local scale. Generally, higher elevations
receive more precipitation and experience lower temperatures and higher wind speeds. In the case
of IAs at high elevations, with Fig. 3, we showed that they receive more precipitation inputs and
experience lower air temperatures. Hence regions at higher elevations, especially above the ELA,
should favour more accumulation than ablation. However, wind-driven snow at higher elevations
does not readily accumulate on steep slopes. Some IAs may take advantage of the leeward
conditions at lower elevations and sustain for more extended periods. Similar results for large



glaciers have previously been reported by Bhambri et al. (2011) or Pandey and Venkataraman

(2013).


**Mean slope:**
Slope angle strongly influences ice velocities of glaciers, mass flux, and the hydrology of the
mountain environments. Its influence on avalanche transport of snow over the glacier surface has
been discussed previously (*e.g.*, Oerlemans, 1989; Hoelzle et al., 2003; DeBeer and Sharp, 2009).
Numerous studies have also reported that slope is the single most crucial terrain parameter that
controls glacier responses to climate change (Furbish and Andrews, 1984; Oerlemans et al., 1998;
Jiskoot et al., 2009; Scherler et al., 2011). Terrain slope has a strong influence on the accumulation
rates in rugged terrains. On steep slopes, accumulation occurring in the temperature range of -5 -
$0^{o}$C can accumulate on steep slopes. Slope likewise plays a key role when calculating other terrain
parameters and indices.

**Mean annual rock surface temperature:**
MARST estimates the average annual temperature of the rock surface governed mainly by the
incoming shortwave solar radiation (PISR) and the mean annual air temperature (MAAT). The
method for estimating MARST is described by Boeckli et al. (2012) and Magnin et al. (2019). The
estimation is based on a multiple linear regression model with the form:

$$Y = \alpha + \sum_{i=1}^{k} \theta i X^{i} + \varepsilon, \tag{5}$$


where Y is the value for MARST, α is the intercept term, $\theta i X^{i}$ represents the model's k variables
(PISR and MAAT) and their respective coefficients, and $\varepsilon$ residual error term distributed equally
with the mean equal to 0 and the variance  $\sigma^{2}$> 0. For predicting the values of MARST in steep
slopes, we use the equation:

$$\text{MARST}_{(pred)} = \alpha + \text{PISR} * b + \text{MAAT} * c, \tag{6}$$




where α is the MARST$_{pred}$ value when PISR and MAAT are equal to 0, and b and c are the
respective coefficients of PISR and MAAT at measured RST positions. These coefficients were
calibrated by Boeckli et al. (2012) (rock model 2) for the entire European Alps using a set of 53
MARST measurement points. The MAAT of the 1961-1990 period was used to calculate MARST,
representing a steady state.
The values for MARST are calculated in ºC and, for our study region, range from -12 to 10ºC.
MARST is also an important criterion to check for the very likely presence of permafrost below
the IAs, which likely allows the formation and existence of IAs.

**Topographic Ruggedness Index:**
The topographic Ruggedness Index (TRI) measures the ruggedness of the landscape. TRI was
calculated based on the methodology proposed by Sappington et al. (2007). It is calculated as a 3-
dimensional dispersion of vectors (x, y, z components) normal to the grid cells considering the
slope and aspect of the cell. The magnitude of the resultant vector in a standardized form (vector
strength divided by the number of cells in the neighbourhood) measures the ruggedness of the
landscape. Higher values of TRI thus suggest a more rugged and sporadic terrain, which could
block the downward movement of the snow and subsequently lead to the formation of a consistent
weak layer, which can destabilize the snowpack and lead to small avalanches resulting in mass
wasting (Schweizer, 2003). Since IA surfaces are smooth, the TRI values calculated at the surface
of the IA is always low. Hence, we consider the TRI values by taking a buffer of 20 m around the
IA boundary delineated for the first observation (1952). The mean TRI value from this buffer is
considered for our analysis.

**Curvature:**
Curvature, estimated as a second derivative of the surface, defines the shape of the slope. Curvature
is considered an essential factor because it can define accumulation or ablation rates for a surface.
It is also considered a significant contributory factor for avalanches (Snehmani et al., 2014).
Generally, two types of curvature profiles are known, plan and profile. For our analysis, we only
used the profile curvature as it defines the shape of the slope in the steepest direction. From a
theoretical point of view, erosion processes prevail in convex (negative values) profile curvature
locations, while deposition is predominant in concave (positive values) profile curvature locations.





The values for the curvature define how strongly convex (lower negative values) or concave
(higher positive values) the slope is. That is why curvatures can be considered an essential role in
the accumulation and ablation rates of a glacier or ice body. Like TRI, the IAs tend to show flat
curvature profiles if we consider their surface. Hence, we estimate the curvature values around the
same buffer as the TRI and use this for further analysis.


**402    4.3 Proxies for ablation and accumulation**


To eventually correlate changes in surface area of IAs with the changing climate, we use the
temperature and precipitation data from the transformed AdM weather records and SAFRAN
reanalysis product (see Sect. 3.1.3) to build proxies for accumulation and ablation. The proxy for
ablation was built by estimating the annual sum of positive degree-days (PDD), computed from
the normal probability distribution centred around the mean monthly temperature.  Estimation of
the PDD is based on the empirical relation, which states that the melting rate is proportional to the
surface-air temperature excess above 0°C. Several methods for estimating the PDD have been
proposed by Braithwaite and Olesen (1989), Braithwaite (1995), and Hock (2003). However, the
method proposed by Calov and Greve (2005) also accounts for stochastic variations in temperature
during the computation of PDD. The formula for the estimation of the PDD using this method is
given by:

$$\text{PDD} = \int_0^A dt \left[ \frac{T_{ac}^2}{\sqrt{2\pi}} \exp\left(-\frac{T_{ac}}{2\sigma^2}\right) + \frac{T_{ac}}{2} \; erfc\left(-\frac{T_{ac}}{\sqrt{2}\sigma}\right) \right] \qquad (7)$$

$T_{ac}$ is the annual temperature cycle (in °C), σ is the standard deviation of the temperature from the
annual cycle, A = 1 year, and *erfc* is the conventional error function built-in in all programming
languages.
After computing the PDD, we calculate the cumulative PDD (CPDD) by taking the sum of all the
annual PDD values for each observation period (*i.e.* 1952-2001, 2001-2012 and 2012-2019). This
value of CPDD is then used as a proxy for ablation (Braithwaite and Olesen, 1989; Vincent and
Vallon, 1997).



The calculation of the proxy for accumulation is more tricky because we only consider the yearly
sum of precipitation occurring at a temperature between -5 and $0^o$ C, as only snowfall within this
temperature range is believed to accumulate/adhere to steep slopes (Kuroiwa et al., 1967; Guillet
and Ravanel, 2020; Eidevåg et al., 2022). The temperature-dependent indicator function can be
written in the following form:

$$\chi_i(T, (t)) = \begin{cases} 1 \text{ if} -5°C \le T(t) \le 0°C \\ \\ 0 \text{ otherwise} \end{cases} \qquad (8)$$


**4.4 Surface area model**

Using the proxy for ablation and accumulation, Guillet and Ravanel (2020) proposed a surface
area model to estimate the differences in the surface areas of IAs between different time steps due
to the time-integrated changes in climate forcing parameters. The main goal is to look for a
potential linear relationship between climate variables and the changes in surface areas of IAs,
using a multivariate regression model. The equation for the model can be written as:

$$S_m(t) = S(t_0) - \int_{t_o}^{t} (\alpha 1\, CPDD(t) - \chi_i(T(t))\alpha 2\, A(t))dt + \beta + \varepsilon(t) \qquad (9)$$

where $S_m(t)$ corresponds to the modelled surface area at time t; similarly, CPDD(t) and A(t)
represent the proxies for ablation and accumulation; S(t = 0) is the first available measurement; α1
and α2 are the coefficients of linear regression, β is the intercept, and ϵ the residual. χ(T, t) accounts
for precipitation occurring in the [-5°C, 0°C] temperature range and is given by the temperature-
dependent indicator function given in equation (5). The area of IAs at each time step was calculated
using the surface area model (with the temperature and precipitation proxies), and we hereafter
refer to this area as modelled area. The measured area is the surface area we delineated using the
high-resolution optical images.

**4.5 Uncertainty estimations**



Since this study uses data from different sources and periods, uncertainties of different origins might have been introduced to delineate the IA boundaries. A good estimation of these uncertainties is thus crucial to have a fair estimation of the significance of the results (Racoviteanu et al., 2008; Shukla and Qadir, 2016; Garg et al., 2017). Some sources of uncertainty in this study could arise from (1) errors inherent to the aerial images and satellite-derived datasets, (2) errors resulting from inaccurate co-registration of data from various sources, (3) errors produced while generating the high-resolution DEM from stereo images, and (4) conceptual errors linked with defining the boundaries of IAs in all images. Quantifying the errors inherent to the processing of all datasets used is challenging, and this is out of the scope of this paper. A detailed accuracy assessment of the DEM generation and co-registration process is provided in Sect. 5.1 and 3.2, respectively. The quantification of errors resulting from the manual delineation of IA boundary is also challenging, but we have previous guidelines from Paul et al. (2017) for the quality and consistency assessment of manual delineations.

One way to assess the area uncertainty is to perform multiple digitizations of the same area and calculate the mean area deviation (MAD), taking the first digitization as a reference (Meier et al., 2018). Considering this, the first author performed multiple digitizations (three times) for 50 IAs on images from 1952, 2001, 2012 and 2019, considering different challenges associated with aerial and satellite images like shadow and illumination. The MAD gives an uncertainty estimate in percentage considering multiple digitizations taking the first digitization as a reference. MAD provides a percentage estimate of how the final area calculated varies across multiple digitizations for each polygon. MAD values are affected by the size of the polygon manually digitized. Previously, authors like Paul et al. (2013), Fischer et al. (2014) and Pfeffer et al. (2014) have reported an increase in the uncertainty of manual digitizations with a decrease in the size of the polygons. With this in mind, we also digitized IAs of different sizes ranging from 0.001 km$^2$ to 0.01 km$^2$.

**5 Results and discussions**

**5.1 Accuracy of the DEM**





Figure 6a shows the stable surfaces (after eliminating glacier boundaries, trees, and forests) we
used for our co-registration process and fig. 6b displays the difference in elevation between the
reference DEM and the source DEM before co-registration. Figure 6c presents the results after the
co-registration process considering all the surfaces (stable and non-stable), and fig. 6d shows the
difference considering only the stable areas after masking out non-stable areas using the glacier
boundaries provided by the RGI. The source DEM was translated using the corresponding shift
values x = -5.03 m, y= 6.00 m, and z = 3.22 m
The distribution of errors can be visualized by a histogram of the sampled errors, where the number
of errors (frequency) within certain predefined intervals is plotted (Höhle and Höhle, 2009). Figure
7 shows the histogram of the errors Δh (elevation difference between the reference and source
DEM) in meters for the stable areas. The accuracy estimates before and after the co-registration is
shown by the normalized median absolute deviation (nmad) and the median value calculated
together. As can be seen, the nmad and median values before the co-registration process for stable
areas were 5.16 and -5.06, respectively. After the co-registration process, the value dropped to
1.98 for the nmad and -0.14 for the median value. This suggests a good correlation between the
high-resolution LiDAR DEM used as a reference and the Pleaides DEM we built.

**5.2 Total area loss of ice aprons in the Mont-Blanc massif over seven decades**

The total area of IAs mapped in 1952 was 7.932 km$^2$. It dropped to 5.915 km$^2$ in 2001. The surface
area further dropped to 4.919 km$^2$ in 2012 and then to 4.21 km$^2$ in 2019 (Figure 8). This implies
that from 1952 to 2019, IAs have lost ~47 % of their original area in 67 years. It corresponds to an
average surface area loss of 0.78 km$^2$ per decade. However, the percentage area loss from 1952 to
2001 was ~26 % compared to ~31 % from 2001 to 2019. This is an alarming rate: IAs have lost
more relative areas during the 18 recent years compared to the previous 50 years (before 2001).
The rate of surface area loss is also disconcerting because, compared to the glaciers in the MBM,
the IAs are losing their area at a higher rate (~24 % for glaciers from the end of LIA till 2008,
according to Gardent et al., 2014). The small size of IAs seems to make them more vulnerable to
global warming than large glaciers. Also, the effects of local topography may be more pronounced
in the case of IAs than for large glaciers. Figure 9 shows the MAD values for 50 IAs in 1952, 2001,
2012 and 2019. We did not observe an increase in MAD values with decreasing size of the IAs,





mainly because the number of samples we used is comparatively less than that in the previous
studies. Overall, the mean MAD observed for all years was ± 6.4 %. The MAD for the IAs digitized
on the orthophotos from 1952 was ± 6.68 %, while for 2001, it was ± 7.2 %. The MAD for 2012
and 2019 was ± 6.32 % and ± 5.50 %, respectively.

**5.4 Influence of the local topography and other factors on the area loss of IAs**

Each parameter, as described in Sect. 4.2 was individually regressed with the relative area loss of
IAs for the three periods, and their influence was assessed by the coefficient of determination ($R^2$)
and Pearson's r-value.
A joint analysis of the surface area lost by the IAs and the direct solar radiation reveals a strong
correlation between the values of DSR and the relative surface area loss of IAs for all the three-
time periods (1952-2001, 2001-2012 and 2012-2019) (Fig. 10a; Table 2). IAs that receive more
radiation from the sun throughout the year lose their surface area faster than those that receive less
DSR (Oerlemans and Klok (2002); Mölg (2004); Johnson and Rupper (2020)). Incoming solar
radiation is also an essential component of all surface energy and mass balance models. However,
this is the first evidence of the potential negative impact of solar radiation on small ice bodies like
IAs. Our previous analysis with the climate variables in Sect. 3.3 indicated a potential relationship
between the elevation and the surface area loss of IAs. This is somewhat statistically significant
from the regression analysis, as we found a negative correlation between the surface area loss and
the mean elevation of the IAs (Fig. 10b; Table 2). IAs present at lower elevations are potentially
subject to intense degradation and lose their surface area faster than those at higher elevations. The
correlation is not particularly strong since, on a more local scale, other topographic factors also
play a critical role in the surface area variations of IAs. However, elevation seems to be particularly
the single dominant causative factor compared to other topographic factors in affecting the
behaviour of IAs to the changing climate. Elevation strongly influences climatic conditions
(temperature, precipitation, and wind speeds) and permafrost; this likely strongly influences the
durability of IAs in the context of changing climate. Hantel et al. (2012) suggested that the median
summer snowline for the Alps to be at 3083 ± 121 m a.s.l. (1961 – 2010), while Rabatel et al.
(2013) documented the regional ELA at 3035 ± 120 m a.s.l. (1984 – 2010). Previous authors further
described the rising of the ELA to 3250 ± 135 m a.s.l. during the 2003 heatwave. Subsequent



heatwaves of 2006, 2015 and 2019 would have likely resulted in similar scenarios (Hoy et al.,
2017). Since 87.5 % of the total IAs mapped (423 in total) exist in elevations above 3100 m a.s.l.
(Kaushik et al., 2021), the rising of the ELA in future climate scenarios leads to more IAs falling
at risk of fast degradation and disappearance. An example of this is the case of the IA on the north
face of Aiguille des Grands Charmoz (3445 m a.s.l.), which completely disappeared during the
2017 summer heatwave (Guillet and Ravanel, 2020).
In addition, we found a moderate positive correlation between the average MARST values and the
surface area loss of IAs. The correlation observed was not very significant compared to the
previous two factors. It indicates that the effect of rock surface temperatures on the area loss of
IAs is not strong on a regional scale, but this could still prove significant on a more local scale.
(Fig. 10c; Table 2). However, this relationship needs to be examined in a more site-specific and
localized area as it was not exposed glaringly for the large sample size and distributed dataset used
for our analysis. We also observed that the correlation degree was higher for a more extensive
observation period (1952-2001) than for shorter periods. This could suggest that the influence of
rock surface temperatures potentially becomes more prominent with a more extensive observation
period. As suggested by Guillet et al. (2021), IAs are cold ice bodies that exist predominantly on
permafrost-affected rock walls. They further reported temperatures <0°C at the base of the ice core
taken from the IA on the north face of Triangle du Tacul (3970 m a.s.l.). Heating from rock surfaces
is predominantly the cause of permafrost degradation, which further affects mountain slope
stability leading to an increased rock mass wasting (Magnin et al., 2017). Cold surfaces
demonstrate more ice cohesion with the underlying surfaces, while a rise in surface temperatures
decreases basal cohesion,  increasing the sliding process and leading to more ice flow (Deline et
al., 2015). Thus, it is likely that underlying permafrost conditions aid the sustainability of IAs in
the long term, and an increase in rock surface temperatures around IAs could result in IAs losing
mass more rapidly.


Kaushik et al. (2021) further showed that most IAs exist in extremely rugged terrains: 51 % of the
total IAs mapped exist in the TRI's high and very high ruggedness class, while only 8 % exist in
the low ruggedness. Thus, comparing the terrain ruggedness to the area loss of IAs makes sense
since the topography around the snow/ice bodies can critically influence their stability (Deline et





al., 2015). Increasing terrain ruggedness is associated with slope instability and further ice volume
loss. However, a similar analysis of IA area loss with the TRI showed a weak positive correlation
(Fig. 10d; Table 2). An increase in TRI values (*i.e.* increase in terrain ruggedness) may result in
more ice area loss on a site-specific scale, but this relationship is hard to observe globally. Like
the results from the analysis with MARST, the strongest correlation was again observed for the
largest study period.

Further, like the TRI, we also found a weak correlation of the terrain slope and curvature with the
surface area loss of IAs. It is important to note that our criteria for selecting IAs already limit us
to areas with slope angles steeper than 40º (Fig.10e; Table 2). Previous analysis (Kaushik et al.,
2021) shows that most IAs in the MBM (83 %) lie at mean slopes between 40° and 65º. Increasing
slope steepness limits accumulation, while avalanches further scour away snow from the surface
of the IA, thus exposing the ice directly to the sun and wind (Vionnet et al., 2012). However, the
differences in slope angles of the IAs was not a dominant factor affecting the rates of area loss. A
plausible explanation for this could be that since we limit the slope criteria to more >40° and most
IAs lie in the range of 40 to 65 º slope angles, the effect of terrain slope is not as well pronounced
as it could be between low (< 15 º) and extreme slopes (>65 º). Similar results were observed by
Li et al. (2011), as they observed very slight variations in area loss for small glaciers with
differences in slope. They suggested other local topographic factors could mitigate the effects of
slope in case of small ice/snow bodies.
Similarly, terrain curvature seems to have the most negligible impact (Fig. 10f; Table 2). As cited
in Sect. 4.2. previous studies may have shown that terrain curvatures could play an essential role
in the dynamics of glaciers, but this is not the case for IAs in the MBM.
We performed the last comparison between the relative surface area loss of IAs with their initial
area. Our results were similar to the one reported by Lopez et al. (2009) as we did not find any
correlation between the two quantities (Fig. 10g; Table 2). Although previous studies by Paul et
al. (2004), Jiskoot et al. (2009), and Garg et al. (2017) have shown the correlation between the size
of the ice/glacier bodies with the area loss, this is not evident in our case. Unlike previous studies,
which considered different glaciers ranging in size from less than a km$^2$ up to several hundred km$^2$,
IAs are small ice bodies (0.0005 km$^2$ to 0.2 km$^2$). Hence, it is plausible that the effect of IA size
related to area loss rate is not pronounced in our case. Similar results were shown by Lopez et al.





(2010), who analyzed 72 glaciers in South America, and reported no correlation between the
glacier length and the area loss of glaciers.


**5.5 Influence of changing climate on the area loss**

Figure 11 shows how the PDD increased over the years in the MBM. All elevations, except 4800
m a.s.l., show an increasing trend of PDD values from 1952 to 2019. Figure 12a presents the
correlation between the ratio of the mean measured surface area at time t, S(t), to the initial area,
S(t$_0$), with the ratio of the mean modelled surface area using the AdM transformed data to the
initial area for 2001 and 1952. We consider the ratio of S(t)/S(t$_0$) as an indicator to estimate the
area loss between the two time periods. A high value of the ratio (*i.e.* value close to 1) in the present
context indicates that the relative surface area loss of IAs between the two periods is comparatively
less than that of IAs whose ratio is closer to 0. A value larger than 1 indicates an increase in the
surface area over time.
From the results, we do not see a strong correlation (r = 0.73) between the modelled area (from
AdM transformed climate data) and the measured area for the 200 IAs spread across the MBM
(Fig. 12a). However, the correlation improves significantly (r = 0.86) when we use the SAFRAN
data based on different elevations and remodel the surface area for each IA (Fig. 12b). This can be
seen from the values of R$^2$, Pearson's r, RMSE and the p-value estimates from the T-test achieved
from both datasets (Table 3). The best-fitting line presents a slope of 1.0 and an intercept of 0.0.
Both figures show that IAs at lower elevations (blue to green colour and small tick size) generally
show lower ratios values than IAs at higher elevations (yellowish colours and bigger tick size).
This implies that the elevation of the IAs potentially plays a crucial role in their response to the
changing climate. Overall, the surface area of IAs has decreased throughout the massif from 1952
to 2001 except for 4 IAs, which showed an increase in surface area. All these 4 IAs are located at
the highest elevation band, which could favour the accumulation and growth of the IAs. The
results, however, reaffirm the proficiency of the proposed surface area model in predicting new IA
states from the accumulation and ablation proxies. Similar results were observed for the other two
time periods, *i.e.* 2001-2012 and 2012-2019, as seen in Table 3.



Our results indicate the strong influence of temperature and precipitation on the surface area
changes of IAs. The results raise further questions regarding the sensitivity of the IAs to extreme
weather events. Unfortunately, our sampling rate does not allow us to quantify the effects of
individual extreme weather events. Nevertheless, there is a strong argument in favour that these
events, especially in the past two decades, cause the IAs to lose mass more rapidly than in the
previous decades. As suggested by Meehl and Tebaldi, (2004), with an increase in the intensity
and frequency of extreme events in the coming decades, understanding the effects of climate
variables on the sensitivity of IAs is even more critical.
Further, several authors have previously also accounted for the variations in solar radiation in
mass-balance modelling studies (Huss et al., 2009; Thibert et al., 2018). Our results showed a
strong correlation of DSR with area change, making this argument stronger. However, since the
focus was to show the impact of climate variables separately, we preferred a temperature-index
model as a first approach. However, we expect solar radiation to play a significant role in the
sensitivity of ice aprons, and future studies on ice apron evolutions in the 21[st] century should
address this question.


**6. Conclusions**

This study makes the first attempt to understand the dynamics of IAs concerning the changing
climate and topographic factors at a regional scale. IAs are very small ice features but highly
critical components of the mountain cryosphere. Because of the difficulties associated with their
monitoring and relative unimportance to mountain hydrology, no studies solely based on their
evolution on a regional scale have been performed before. This paper presented an analysis of 200
IAs spread throughout the MBM and existing in different topographic settings better to understand
their dynamics in the context of climate warming. For this purpose, we accurately mapped the IAs
on very high-resolution aerial and satellite images available for 1952, 2001, 2012 and 2019. Using
our extensive database of IAs, we compared the total area variation of IAs for three periods.
Further, we also attempted to establish a relationship between the surface area lost by IAs with
climate forcing parameters (*i.e.* temperature and precipitation) and their associated topographic
parameters.




Some important outcomes are:
• Over the study period 1952-2019, IAs have lost their surface area at a very alarming rate.

The total area of IAs in MBM was 7.93 km$^2$ in 1952, which dropped to 5.91 km$^2$ in 2001,

to 4.91 km$^2$ in 2012, and at last to 4.21 km$^2$ in 2019 (~ 47% drop in total surface area in

less than three-quarters of a century).

• The observed rate of relative area loss in the last 18 years (~31 %) is more than the overall

area loss during the 48 previous years (1952-2001; ~ 26 %).

• Results from the analysis of IA surface area loss and climate forcing parameters

conclusively proved the strong impact of climate forcing parameters on the behaviour of

small ice bodies like IAs.

• Further analysis of IA surface area loss with different topographic parameters showed

possible strong links of some topographic factors with the area loss of IAs, while other

factors are not relevant, at least on a regional scale.

• The strongest correlation of IAs surface area loss was found with the DSR and elevation.

Other factors like MARST, TRI, and mean terrain slope could also play an important role

locally, but their effect is not significant regionally. Other factors like terrain curvature and

the size of the IAs were not found to impact the IA's surface area loss significantly.


Looking at the melting rate of IAs and the future predictions of global climate change, it is evident
that these small and critical ice bodies are most vulnerable to adverse impacts. It is hard to imagine
any of the IAs surviving the next few decades with increasing temperatures at the present and
future melting rates. The loss of IAs will thus be the loss of crucial glacial heritages and
playgrounds for the iconic practice of mountaineering. Hopefully, this study forms a basis to
encourage further studies on IAs.


**Author contributions:** SK designed the study and drafted the paper, which was revised by all co-
authors. LR and FM helped in data interpretation and analysis. YY and ET proofread the
manuscript and provided valuable inputs for improving the overall quality of the paper. DC
processed and provided the DEM used for the study.




**Data availability:** The ice apron inventory will be made available on demand.

**Competing Interests:** The authors declare that they have no conflict of interest.

**Acknowledgements:**
This research is part of the USMB *Couv2Glas* and *GPClim* projects. Pleiades data were acquired
within the CNES *Kalideos-Alpes* project and successfully processed under the program "*Emerging*
*risks related to the 'dark side' Alpine cryosphere*". We also thank C. Vincent of the *Institut des*
*Géosciences de l'Environnement* (IGE) for providing the LiDAR DEM os the Argentière glacier
area.

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





**Tables:**

| Data type | Source | Resolution (m/time) | Acquisition time/period |
|---|---|---|---|
| Optical | Orthoimages IGN | 0.2 | July 2015 |
| | Pleiades 1A PAN | 0.5 | 25/08/2019, 19/08/2012 |
| | Sentinel 2 | 10 | 12/09/2019 |
| | SPOT 6 | 2.2 | 14/09/2019 |
| | Pleiades 1A XS | 2 | 19/08/2012 |
| | Orthoimages IGN | 0.5 | July 2001 |
| | Orthoimages IGN | 0.5 | 1952 |
| | SAFRAN reanalysis | daily | 1958 - 2019 |
| Meteorological | Col du Grand-Saint Bernard weather station (2469 m a.s.l.) | daily | 1952 - 2019 |
| | Aiguille du Midi weather station (3840 m a.s.l.) | daily | 2007-2018 |
| | | | |

Table 1: Datasets used for the study.

| Variable | Time-period | $R^2$ | Pearson's r |
|---|---|---|---|
| **Direct Solar Radiation** | 1952 - 2001 | 0.64 | 0.79 |
| | 2001 - 2012 | 0.67 | 0.81 |
| | 2012 - 2019 | 0.51 | 0.72 |
| | 1952 – 2001 | 0.61 | -0.78 |
| | 2001 - 2012 | 0.57 | -0.75 |
| **Elevation** | 2012 - 2019 | 0.51 | -0.71 |
| | 1952 - 2001 | 0.40 | 0.63 |
| | 2001 - 2012 | 0.34 | 0.58 |





| | | | |
|---|---|---|---|
| **MARST** | 2012 - 2019 | 0.27 | 0.52 |
| **TRI** | 1952 - 2001 | 0.37 | 0.60 |
| | 2001 - 2012 | 0.30 | 0.55 |
| | 2012 - 2019 | 0.32 | 0.57 |
| **Slope** | 1952 - 2001 | 0.29 | 0.54 |
| | 2001 - 2012 | 0.25 | 0.50 |
| | 2012 - 2019 | 0.21 | 0.46 |
| **Curvature** | 1952 - 2001 | 0.06 | -0.26 |
| | 2001 - 2012 | 0.03 | -0.18 |
| | 2012 - 2019 | 0.06 | -0.24 |
| **Size of IA** | 1952 - 2001 | 0.04 | -0.22 |
| | 2001 - 2012 | 0.06 | -0.26 |
| | 2012 - 2019 | 0.04 | -0.22 |

Table 2: Linear regression parameters and correlation metrics for each studied parameter.



| Time period | Slope | Intercept | $R^2$ | Pearson's r | RMSE ($km^2$) | p value |
|---|---|---|---|---|---|---|
| 1952 - 2001 | 0.79 | 0.12 | 0.53 | 0.73 | 0.010 | < 0.001 |
| 2001 - 2012 | 0.70 | 0.26 | 0.56 | 0.75 | 0.102 | < 0.001 |
| 2012 - 2019 | 0.89 | 0.04 | 0.63 | 0.80 | 0.097 | < 0.001 |
| | | | | | | |

(a)


| Time period | Slope | Intercept | $R^2$ | Pearson's r | RMSE ($km^2$) | p value |
|---|---|---|---|---|---|---|
| **1952 - 2001** | 1.01 | -0.04 | 0.73 | 0.86 | 0.075 | < 0.001 |
| **2001 - 2012** | 0.74 | 0.22 | 0.67 | 0.82 | 0.086 | < 0.001 |





| **2012 - 2019** | 1.37 | -0.39 | 0.83 | 0.91 | 0.071 | < 0.001 |
|---|---|---|---|---|---|---|

(b)
Table 3: Linear regression parameters and correlation metrics for each time-period (a) using
AdM transformed data and (b) using the SAFRAN reanalysis product.


























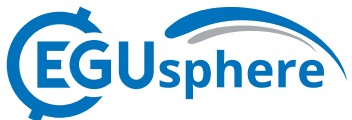

**Figures:**

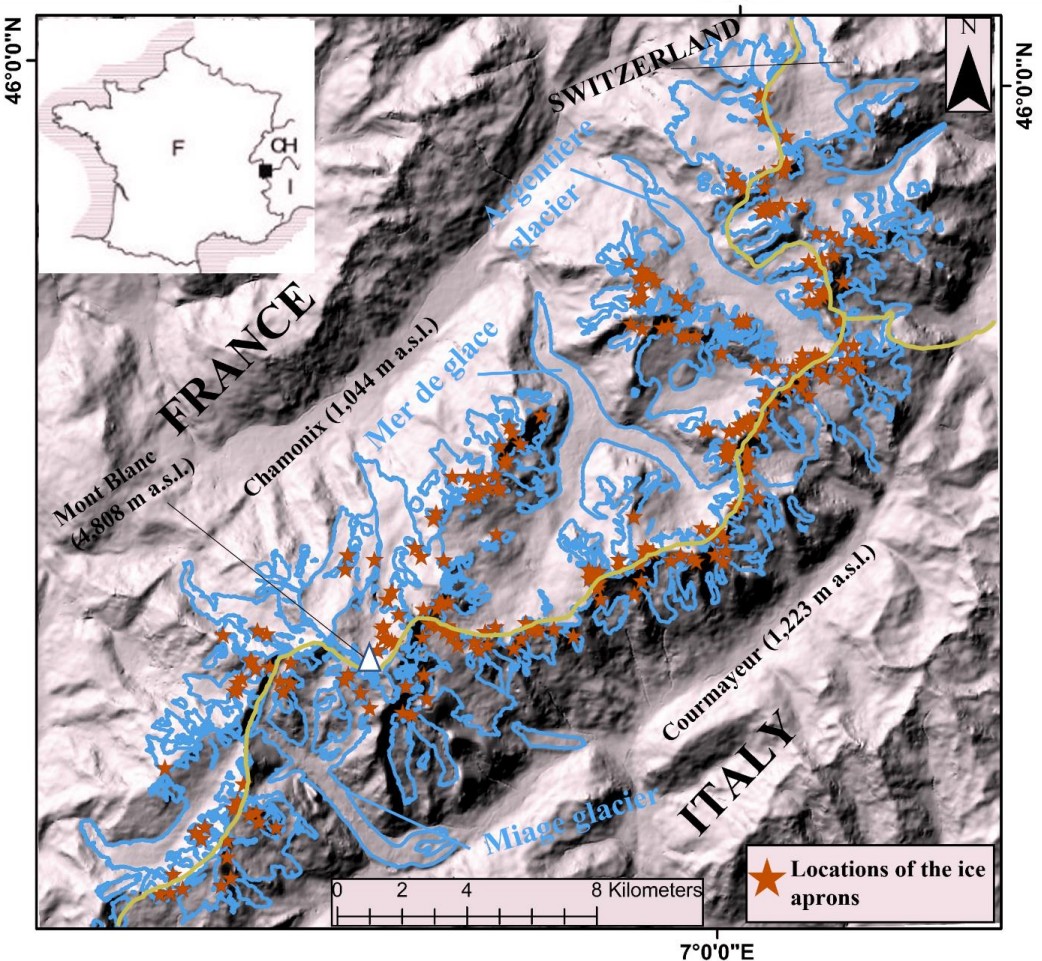


Figure 1: The Mont-Blanc massif (Western European Alps). 200 IAs (red stars) were digitized
accurately on high-resolution images. The glacier outlines (in blue) comes from Gardent et al.,
2014.





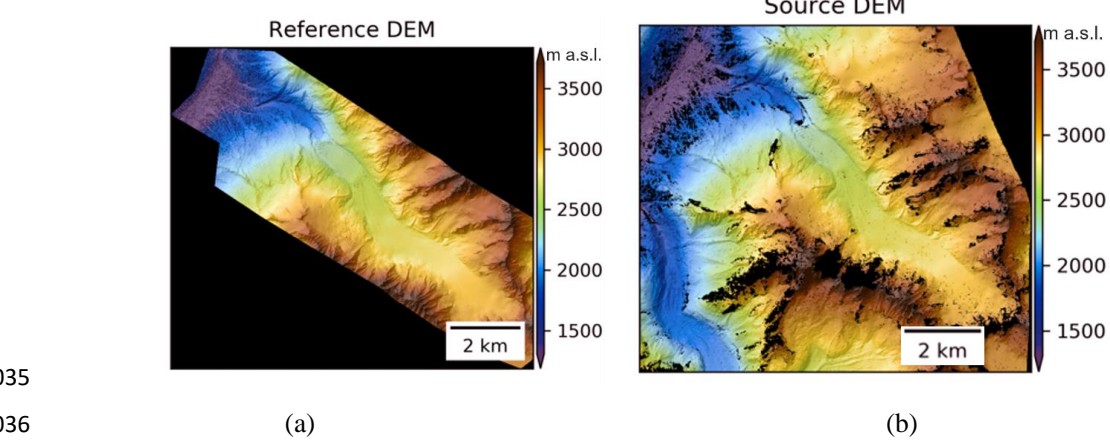


(a)                                (b)

Figure 2: (a) The reference LiDAR DEM of the Argentière glacier used for co-registration, (b)
the source Pleiades DEM used for further analysis.

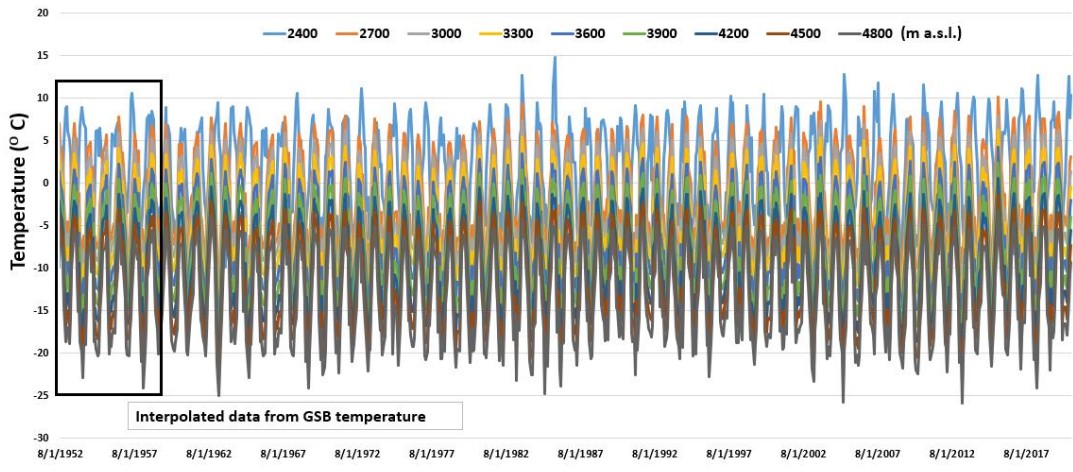


Figure 3: SAFRAN reanalysis product temperature time-series from 1952-2019 for different
elevations in the MBM. The figure shows mean August temperatures, the peak summer month in
the Alps.








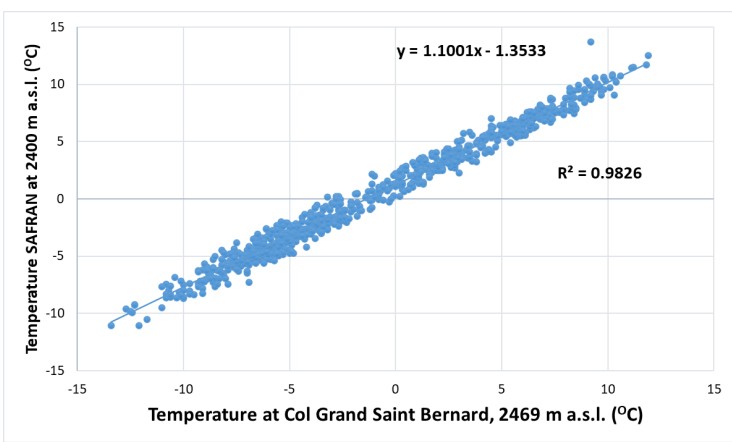


Figure 4: Correlation between the monthly averaged temperature measurements at the Col du Grand Saint Bernard (GSB) and the SAFRAN reanalysis data at 2400 m a.s.l.




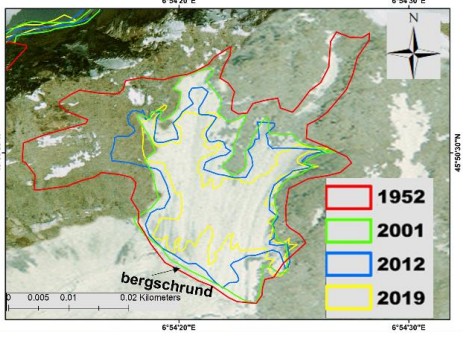


(a)                                                                (b)










(c)                                                    (d)

Figure 5: IAs extent delineated on high-resolution images: (a) orthophotos 1952, (b) orthophotos 2001, (c) Pleaides panchromatic 2012, (d) Pleaides panchromatic 2019. The different colour polygons represent the surface area for each date.

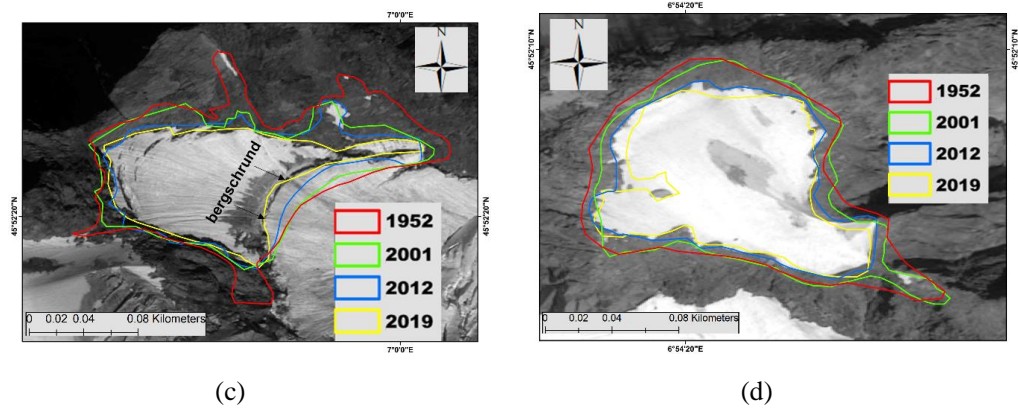

(a)                                                    (b)





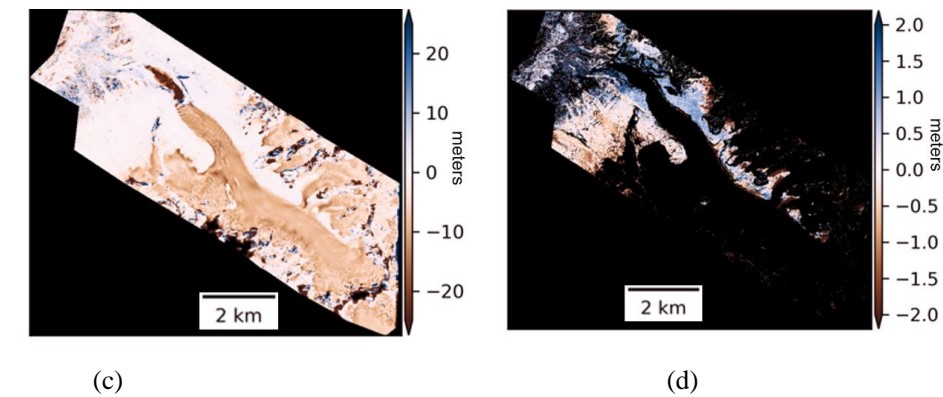


1072      (c)             (d)


Figure 6: Stepwise Pleiades DEM accuracy assessment (a) the surfaces used for coregistration
(b) elevation difference before coregistration (c) elevation difference after coregistration
considering all areas (d) elevation difference after coregistration considering only the stable
areas.

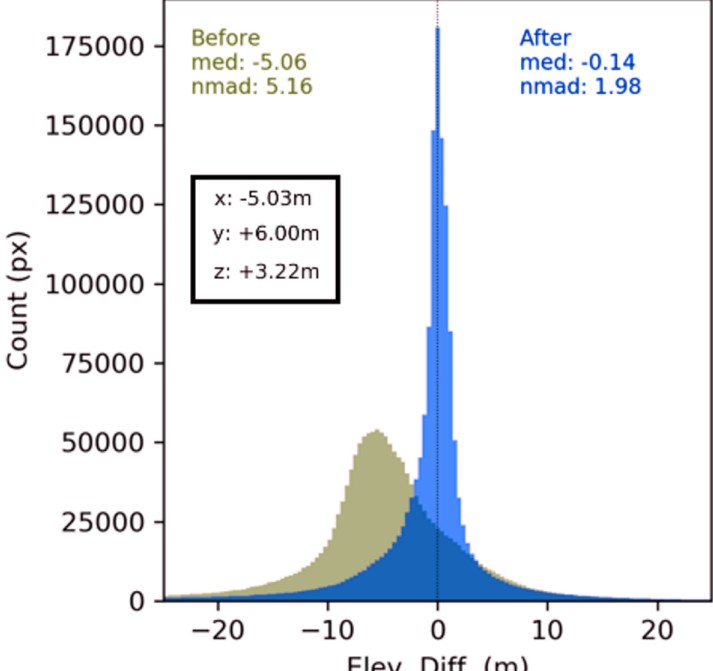



Figure 7: DEM Error (elevation difference between the reference and source DEM) distribution
for stable areas







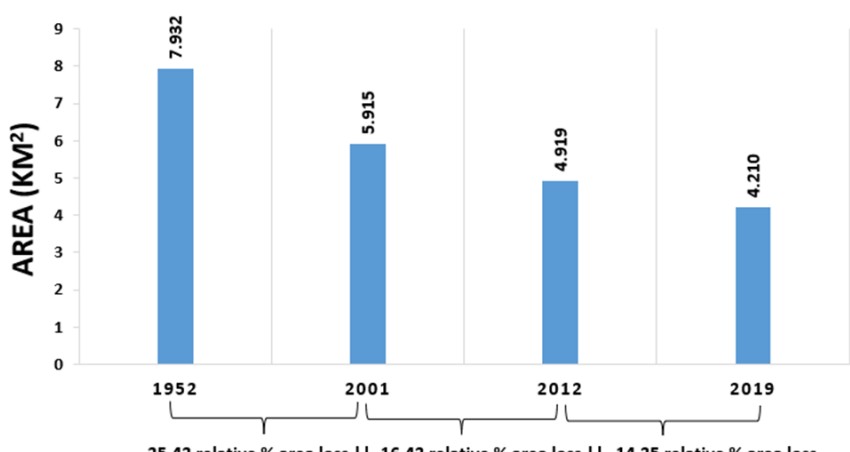


Figure 8: A comparison of the total surface area of all IAs (423 IAs) in the MBM over 67 years.


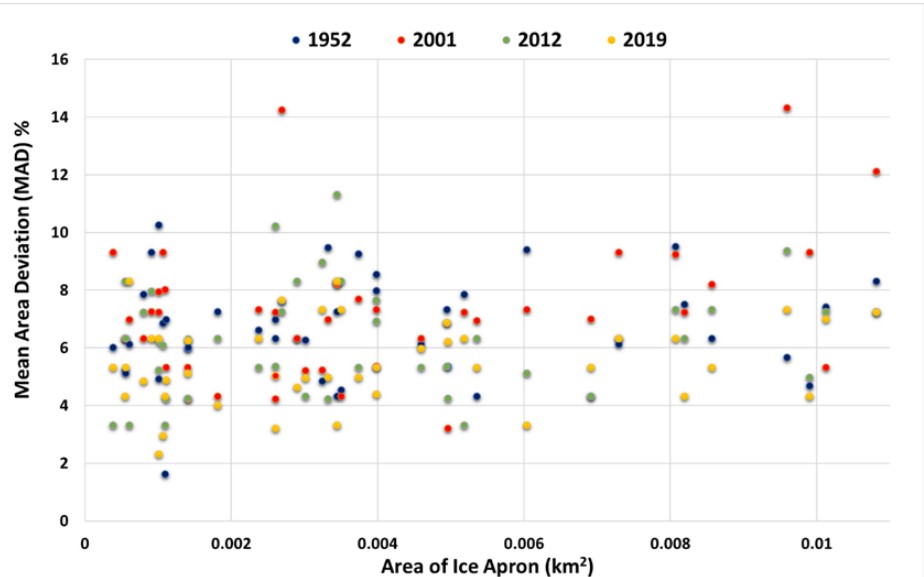


Figure 9: The distribution of MAD values based on multiple digitizations of the IAs area for all periods.






(a)                                                    (b)




(c)                                                    (d)



(e)                                                    (f)





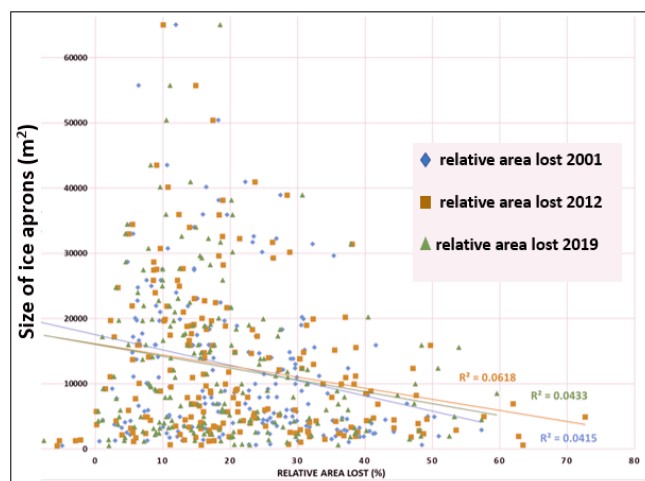

(g)

Figure 10: Scatter plots showing relationships between topographic factors and the area loss of
IAs from 1952 to 2019. a) Direct solar radiation, b) elevation, c) TRI, d) MARST, e) slope, f)
curvature and f) size of the IAs.

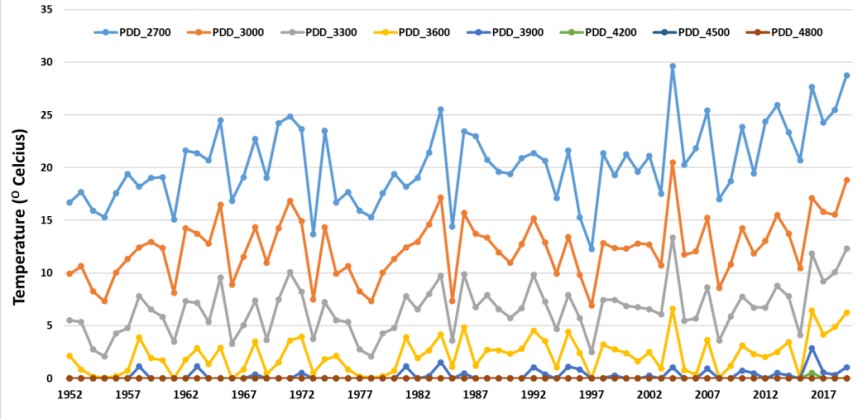

Figure 11: The variation of annual PDD values at different elevations in the MBM from 1952 to
2019.

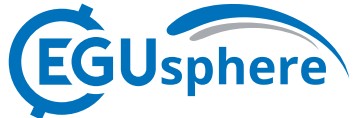


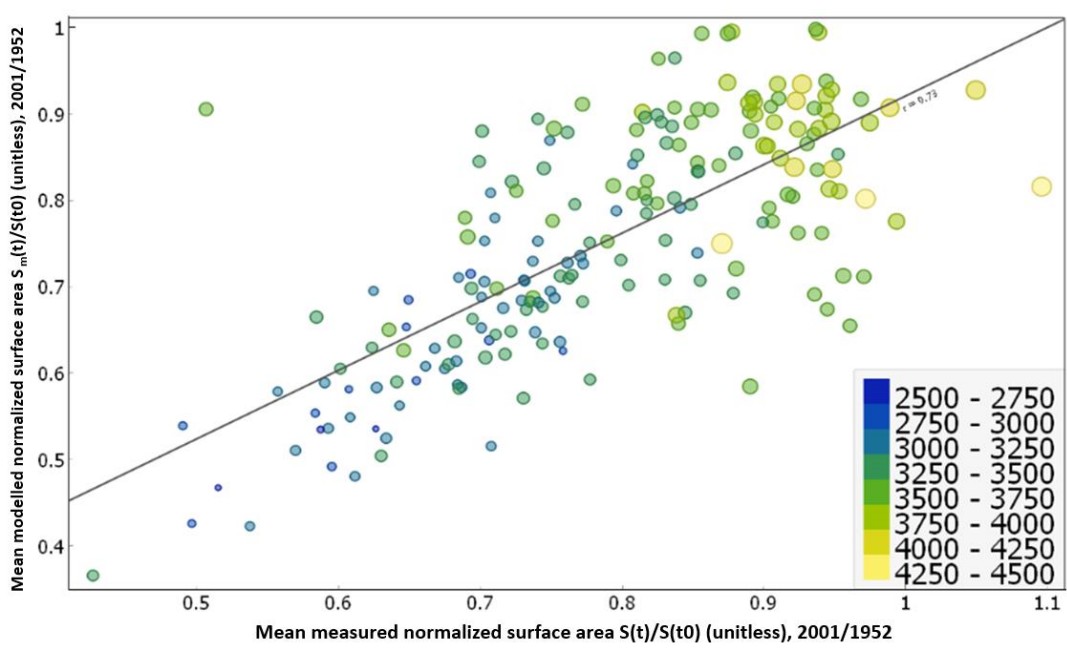


(a)

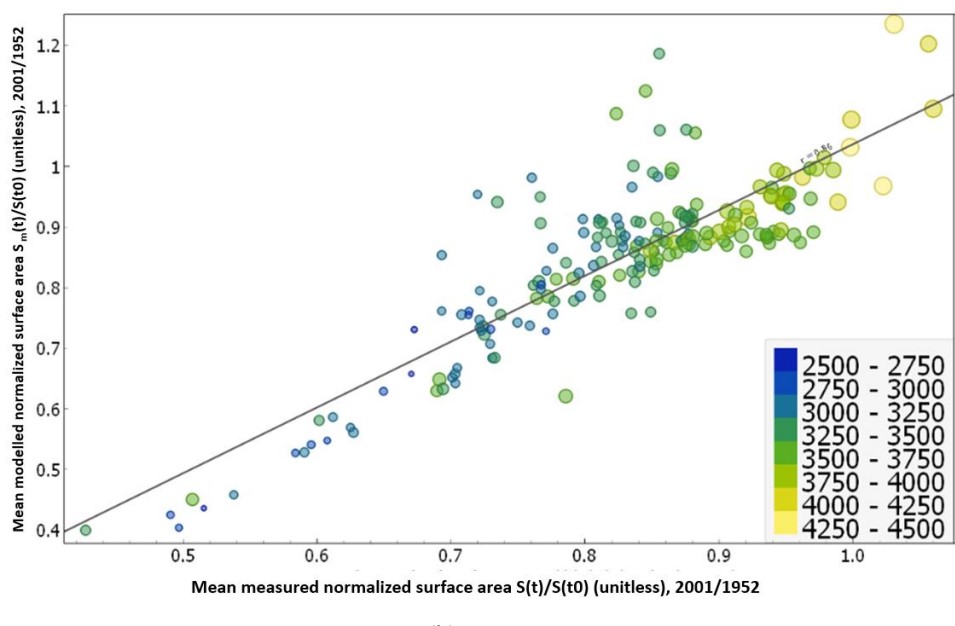


(b)






Figure 12: Correlation between the mean normalized surface area estimates and the modelled
surface areas. $S(t)/S(t_0)$ represents the normalization of the surface area measurement at time t by
the initial value. Similarly, $Sm(t)/S(t_0)$ represents the normalization of the modelled surface area
(a) with GSB data transformed to AdM data and (b) with SAFRAN reanalysis data at time t. The
colour and size of the ticks represent the mean elevation of the IA.
