# Peer review of "Effects of topographic and meteorological parameters on the surface area loss of ice aprons in the Mont-Blanc massif (European Alps)"

_EGUsphere, 2022_

## Referee Comment (RC1)

Review of the manuscript Egusphere-2022-334 "Effects of topographic and meteorological parameters on the surface area loss of ice aprons in the Mont Blanc massif (European Alps)" by Kaushik et al.

**General comments**

The paper from Kaushik et al. presents a study on ice aprons (IAs), very small ice bodies located on the steep slopes of the Mont Blanc massif. They inventoried 200 of these ice bodies at different time intervals (from 1952 to 2019), assessing the changes in their surface area in relation to various topographic parameters and to accumulation and ablation proxies constructed using air temperature and precipitation data.

The paper presents new data and is a valuable contribution to the knowledge of a still unknown component of the Alpine cryosphere.

The main findings of the work are the greater surface loss observed in the last two decades compared to previous periods, the impact of climate forcings on this loss, and the strong correlation between surface loss and parameters such as direct solar radiation and elevation.

The introduction is adequate and well supported by references, also considering that there are no previous studies on this type of ice features.

The used data and methods are well described, and accuracy and uncertainty are properly evaluated.

Results are mostly well presented and sufficient to support the interpretations. However, to improve clarity and linearity, the authors should consider separating the results and discussion into two distinct sections. In some cases, the discussion could be more thorough, considering more adequately alternative hypotheses such as the role of avalanching in the dynamics of the investigated ice bodies (see "specific comments").

I also emphasised the need to carefully separate the results concerning this study from those concerning previous studies conducted by the same authors (see "minor comments").

Figures are adequate in number and generally well made. However, some figures need improvements (see "minor comments").

The two main issues that arose from reading the paper are reported below (specific comments), whereas a number of minor/technical issues are listed in the last section (minor comments/technical corrections).

As an overall assessment, in my opinion the article is suitable for the scope of The Cryosphere and can be considered for publication after making minor revisions.

**Specific comments**

1) Accumulation and ablation processes on IAs can be very complex and, among the factors that determine their existence and evolution, the avalanche contribution has not, in my opinion, been adequately considered and discussed by the authors. The role of avalanches in the dynamics of these very small ice bodies can be crucial, as these processes can accumulate or remove large amount of snow from their surface in a way that is partially decoupled from the regional climatic conditions (i.e., air temperature and precipitation), and strongly related to the local topography and wind.
I'm aware that is rather hard to estimate/model the avalanche activity on IAs, however, the authors can evaluate the use of a simple approach like that proposed by Hughes (2008) and adopted also in Carturan et al., 2013.

2) It seems that in the abstract and along the manuscript the terms "climate forcing parameters", "meteorological parameters" and "climatic factors" are used interchangeably. In my opinion, these terms should be used more properly, because they do not have the same meaning. I suggest the authors to justify why they use these different definitions or, alternatively, be consistent in using only one of them.

References

Hughes, P. D. (2008). Response of a Montenegro glacier to extreme summer heatwaves in 2003 and 2007. *Geografiska Annaler: Series A, Physical Geography*, *90*(4), 259-267.

Carturan, L., Filippi, R., Seppi, R., Gabrielli, P., Notarnicola, C., Bertoldi, L., ... & Dalla Fontana, G. (2013). Area and volume loss of the glaciers in the Ortles-Cevedale group (Eastern Italian Alps): controls and imbalance of the remaining glaciers. *The Cryosphere*, 7(5), 1339-1359.

**Minor comments/technical corrections**

L 47: The terms "advance" and "retreat" are usually referred to the glacier snout; when speaking about glacier volume and area, it is better to use more appropriate terms.

L 55: It seems that the subject here is "variable responses" and not "variable". Please check.

L 112: Use "hosts" or "includes" instead of "displays"; remove "some of".

L 125: Check the elevation of Chamonix, which is not consistent with L 114.

L 131-134: Add a citation here to support these data and quantify the reported threshold.

L 139 – 154: This part could be summarized.

L 157 (section 3): check consistency in the use of verb tenses along this section (present vs past tense); I would prefer the use of past tense.

L 161 (section 3.1): this section could be shortened, as the procedure followed to create and co-register the 2019 DEM from Pleiades images has already been thoroughly described in Kaushik et al. 2021; here, it would be adequate to cite this paper and summarize the description.

L 180-181: could you give here more information on the 2m DEM used for co-registration? (e.g., spatial extent and location of the area used to perform the procedure).

L 182: *outlines* instead of *contours.*

L 188: check whether acronyms were written out in full the first time they were used (e.g., MARST).

L 191 (section 3.2): after reading this section, it is not clear which images were actually used in the study; please, try to be clearer and check consistency with table 1.

L 195: check the consistency of the resolution of 1952 and 2001 images with that reported in table 1.

L 216 (section 3.3): In my opinion, this important section should be clarified, because there is some confusion between what was done in a previous study (i.e., Guillet and Ravanel, 2020) and what was done in this study.

More specifically, L 227 to L 247 refer to the previous study (conducted on 6 IAs), and in particular to the correlation between GSB and AdM, which was used to extend the Adm record over time. In this study (L 248 – L 264), the SAFRAN dataset was used, and a similar correlation with GSB was used to extend this dataset back in time (from 1958 to 1952). Then, the full SAFRAN dataset was extrapolated to the elevation belts considered in this study.

Try to clarify these points by clearly separating what was done in Guillet and Ravanel, 2020 and what was done for this study.

L 231: *located* instead of *present.*

L 278 – 279: the 2001 image is from July, not exactly the end of the ablation season; could a possible overestimation of the IAs extent for that year be considered?

L 312: a parenthesis is missed after the exponent of the second term ($\theta$).

L 299-302: this introductive part could be shortened or omitted.

L 345-346: this sentence is not clear, try to rephrase.

L 393-394: are 'erosion' and 'deposition' here referring to snow or to the avalanche activity? Please clarify.

L 405: why is AdM weather record cited here? From section 3.3 one can understand that GSB record was used along with SAFRAN dataset, at least in this work. See comments for section 3.3 and try to clarify.

L 406: Sect. 3.3?

L 418: please, clarify what does it mean "annual temperature cycle".

L 449: equation (8).

L 465: check the consistency of the section numbers (e.g., DEM generation is described in sec. 3.1).

L 469: …better to use "of the same *surface"* instead of *"area".*

L 473-475: check for some repetitions here. (e.g., L 470).

L 485 (section 5.1): as with the comments in section 3.1, the text could also be summarized here, referring to what is reported in Kaushik et al. 2021, where the procedure is adequately described. In my opinion, the figures referring to this section (Fig. 6 and 7) should also be somewhat differentiated from those of Kaushik et al. 2021; perhaps the comment to the Fig. 6 might help (see below).

Alternatively, this part (section 5.1) could be summarized and included in a section of chapter 4 where an overall estimation of errors and uncertainties are provided, e.g. merged with the section 4.5.

L 535: add reference to Guillet and Ravanel (2020) instead of reference to the section 3.3.

L 538: *located* instead of *present.*

L 548: add a citation for this estimate.

L 550: exist *at…*

L 550: The reader remains a bit confused here. These numbers refer to the whole inventory of IAs (n=423), which is provided in Kaushik et al., 2021. However, only 200 IAs were considered in this work, i.e., a sub-sample of the complete inventory.
I would suggest being clearer, carefully distinguishing here and in the following part of the paragraph what refers to the previous work (with the full inventory) and what refers to the present work.
Are the correlations similar when considering the full inventory and the sub-sample? Are there differences?

L 559: the reference is to Fig. 10d; in general, check the consistency between the text and the sequence of plots in Fig. 10.

L 600-602: I do not find in section 4.2 what is stated here.

L 604: Lopez et al., 2009 not reported in the reference list.

L 609: not consistent with introduction (IAs area smaller than 0.1 km$^2$).

L 610: Lopez et al., 2010 not reported in the reference list.

L 618: show the trends in Fig. 11.

L 665: *in order to* instead of *better to*?

L 680: IA*s.*

L 683: IA*s.*

L 680-682 and 683-685: rephrase these sentences avoiding repetitions.

References (already included in the manuscript)

Kaushik, S., Ravanel, L., Magnin, F., Yan, Y., Trouve, E., and Cusicanqui, D.: DISTRIBUTION AND EVOLUTION OF ICE APRONS IN A CHANGING CLIMATE IN THE MONT-BLANC MASSIF (WESTERN EUROPEAN ALPS), Int. Arch. Photogramm. Remote Sens. Spatial Inf. Sci., XLIII-B3-2021, 469–475, https://doi.org/10.5194/isprs-archives-XLIII-B3-2021-469-2021, 2021.

Guillet, G., & Ravanel, L. (2020). Variations in surface area of six ice aprons in the Mont-Blanc massif since the Little Ice Age. Journal of Glaciology, 66(259), 777-789. doi:10.1017/jog.2020.46

**Comments to figures and captions**

General comment to figures: check consistency between the figures regarding the font used for labels, numbers, and letters, etc.

Fig. 1

I suggest using black squares to indicate the correct location of the two towns shown on the map. (i.e. Chamonix and Courmayeur).

The blue labels are not clearly visible, try changing the colour.

For scale and north arrow: leave some space between the frame and the graphic item.

Fig. 3 and Fig. 4

Some labels should be enlarged to improve readability.

Fig. 5

I suggest combining the four figures into a composite one, using the north arrow and legend only once.

The scale and coordinate labels should be enlarged.

Be consistent with Fig. 1 in the type of north arrow used.

The year of the orthophotos used as background should be reported.

Fig. 6

The same suggestion as Fig. 5: should be combined into a composite figure.

Add north arrow.

Fig. 10

The seven plots should be of the same size.

Some labels should be enlarged, and the size should be the same among the plots.

The legend could only be used once if the graphs were combined in a composite figure.

Fig. 11

Check the label: Temperature (°C).

Some labels should be enlarged.

Fig. 12

The two plots could be combined in a composite figure, using the legend once, the same size of the frames and the same scale for the x-axis.

Figure caption: "Correlation between the mean normalized measured and modelled surface areas".

---

## Author Comment (AC1)

**General comment:**

The paper from Kaushik et al. presents a study on ice aprons (IAs), very small ice bodies located on the steep slopes of the Mont Blanc massif. They inventoried 200 of these ice bodies at different time intervals (from 1952 to 2019), assessing the changes in their surface area in relation to various topographic parameters and to accumulation and ablation proxies constructed using air temperature and precipitation data.

The paper presents new data and is a valuable contribution to the knowledge of a still unknown component of the Alpine cryosphere.

The main findings of the work are the greater surface loss observed in the last two decades compared to previous periods, the impact of climate forcings on this loss, and the strong correlation between surface loss and parameters such as direct solar radiation and elevation.

The introduction is adequate and well supported by references, also considering that there are no previous studies on this type of ice features.

The used data and methods are well described, and accuracy and uncertainty are properly evaluated.

Results are mostly well presented and sufficient to support the interpretations. However, to improve clarity and linearity, the authors should consider separating the results and discussion into two distinct sections. In some cases, the discussion could be more thorough, considering more adequately alternative hypotheses such as the role of avalanching in the dynamics of the investigated ice bodies (see "specific comments").

I also emphasized the need to carefully separate the results concerning this study from those concerning previous studies conducted by the same authors (see "minor comments").

Figures are adequate in number and generally well made. However, some figures need improvements (see "minor comments").

The two main issues that arose from reading the paper are reported below (specific comments), whereas a number of minor/technical issues are listed in the last section (minor comments/technical corrections).

As an overall assessment, in my opinion the article is suitable for the scope of The Cryosphere and can be considered for publication after making minor revisions.

**REPLY:**

Dear reviewer, firstly, we would like to thank you for such a rigorous review, highlighting our manuscript's importance and suggesting valuable inputs to improve the manuscript further. We are sure your comments will improve the quality and readability of our manuscript. We will try to incorporate your suggestions in all possible means to advance our manuscript.

Just to answer one suggestion you have here regarding separating the results and discussion sections to improve the clarity and linearity of the manuscript. We totally agree with this suggestion. We will have a separate discussion and results section in our revised manuscript to improve its readability.

Below you will find attached our reply to your specific and minor/technical comments. We thank you again in advance for reading our replies and also reviewing the revised manuscript if the editor deems it suitable.

**Specific comments:**

**Accumulation and ablation processes on IAs can be very complex and, among the factors that determine their existence and evolution, the avalanche contribution has not, in my opinion, been adequately considered and discussed by the authors. The role of avalanches in the dynamics of these very small ice bodies can be crucial, as these processes can accumulate or remove large amount of snow from their surface in a way that is partially decoupled from the regional climatic conditions (i.e., air temperature and precipitation), and strongly related to the local topography and wind. I'm aware that is rather hard to estimate/model the avalanche activity on IAs, however, the authors can evaluate the use of a simple approach like that proposed by Hughes (2008) and adopted also in Carturan et al., 2013.**

**REPLY:**

- Dear reviewer, thanks a lot for this suggestion. We completely agree with you that the role of avalanches could be crucial in the ablation and accumulation of snow from IAs surfaces; however, we believe this is extremely difficult to estimate/model considering the small size of the IAs. This challenging but interesting topic could form the next topic of our research. Since our paper is already extremely long, with all due respect to your suggestion, we feel this may be out of scope for this article.
We can, however, highlight this point as a potential limitation/shortcoming of our study and something our future research can focus on.

**It seems that in the abstract and along the manuscript the terms "climate forcing parameters", "meteorological parameters" and "climatic factors" are used interchangeably. In my opinion, these terms should be used more properly, because they do not have the same meaning. I suggest the authors to justify why they use these different definitions or, alternatively, be consistent in using only one of them.**

**REPLY:**

Dear reviewer, thank you so much for raising this issue. We agree that the use of different terminologies can be very confusing for the readers. In order to avoid confusion, as suggested by you, we will use just one terminology, i.e. 'meteorological parameters', throughout the manuscript since this term also occurs in the manuscript title. We will also ensure that the terminology is well explained in the introduction to avoid any confusion for the readers.

**Minor comments/technical corrections:**

**L 157 (section 3): check consistency in the use of verb tenses along this section (present vs past tense); I would prefer the use of past tense.**

- Thank you so much for your suggestion. We will rewrite this section in the revised manuscript using only the past tense.

**L 161 (section 3.1): this section could be shortened, as the procedure followed to create and co-register the 2019 DEM from Pleiades images has already been thoroughly described in Kaushik et al. 2021; here, it would be adequate to cite this paper and summarize the description.**

- We agree with your suggestion. We will shorten this section and cite our previously published article to avoid any significant overlaps.

**L 180-181: could you give here more information on the 2m DEM used for co-registration? (e.g., spatial extent and location of the area used to perform the procedure).**

- The LiDAR DEM used for co-registration was acquired for an 8 * 2.5 km area around the Argentiere glacier. We will add this information also to the manuscript.

**L 182: outlines instead of contours.**

**L 188: check whether acronyms were written out in full the first time they were used (e.g., MARST).**

- We will consider both these issues and make necessary changes to the revised manuscript.

**L 191 (section 3.2): after reading this section, it is not clear which images were actually used in the study; please, try to be clearer and check consistency with table 1.**

**L 195: check the consistency of the resolution of 1952 and 2001 images with that reported in table 1.**

- Dear reviewer, for 1952 and 2001, our mapping exercise relied only on the ortho-images for these two periods since high-resolution images from any other source were unavailable. For 2012 and 2019, we had data from multiple sources (Pleiades, SPOT and ortho-images, Sentinel 2), but we utilized a combination of Pleiades and SPOT 6 XS images for mapping the IAs boundaries, with further validation conducted using the orthoimages. We will make this point explicitly clear in our texts for section 3.2. We will further check the consistency of the resolutions mentioned in the text with table 1.

**L 216 (section 3.3): In my opinion, this important section should be clarified, because there is some confusion between what was done in a previous study (i.e., Guillet and Ravanel, 2020) and what was done in this study.**

**More specifically, L 227 to L 247 refer to the previous study (conducted on 6 IAs), and in particular to the correlation between GSB and AdM, which was used to extend the Adm record over time. In this study (L 248 – L 264), the SAFRAN dataset was used, and a similar correlation with GSB was used to extend this dataset back in time (from 1958 to 1952). Then, the full SAFRAN dataset was extrapolated to the elevation belts considered in this study.**

**Try to clarify these points by clearly separating what was done in Guillet and Ravanel, 2020 and what was done for this study.**

- Dear reviewer, after reading this complex section, we agree with our comment that there is a need to separate this section in order to distinctly separate the previous work from our work. We will work on this section and make two separate paragraphs about to clearly define and separate our work from the previous work.

  To clear up your confusion, two datasets were used for our analysis. The first comparison was made using transformed AdM data (the results of which are presented in figure 11a). This is the dataset also used by Guillet et.al 2021 for their analysis. We call it the transformed AdM dataset because data from AdM station is available only from 2007, so a linear relationship was established with the GSB data to extend it in the past. Using this relationship, all the GSB data was converted to AdM data (maybe the correct term here could transformed GSB data rather than transformed AdM data). The limitation of this dataset is that we use data recorded for just one elevation (3840 m a.s.l.) and apply the to built a model for the entire massif. But the spread of our IAs ranges from 2604 to 4500 m a.s.l. Further precipitation values are not available from the AdM weather station, so

Guillet et.al, 2021 used the precipitation values from the GSB station and applied them to build their proxies for the entire MBM. Since the previous study relied only on 6 IAs, the uncertainty in the estimation was most likely not observed.

- However, in our case, we work with a large inventory of 200 IAs spread at different elevations and aspects throughout the massif. This uncertainty could have been significant in our case. So, to avoid this uncertainty, we used the SAFRAN datasets, which provide modelled temperature and precipitation data at every 300 m elevation bands for the entire massif. The problem we encountered with the SAFRAN dataset was that this dataset starts from 1958, but our first observation date was in 1952. So in order to extend our datasets, we also used the GSB data and built a linear relationship with the SAFRAN data (we used the SAFRAN temperature modelled at 2400 m a.s.l to match the elevation of GSB station). After this, we transformed our SAFRAN data using this linear relationship for IAs at 2400 m a.s.l., and then used an elevation gradient of -0.53 oC/100 m. This figure was taken from previous studies by Magnin et.al, 2015. Precipitation does not show any linear trend, so we used the same precipitation values as taken from the GSB data for all the elevation bands from 1952 to 1958.

Figure 11 a and b show the improvement in the model-based estimate between both methods, where we observed an r-value of 0.73 with the first data (used by Guillet et al.,2021) and an r-value of 0.86 with our data and method.

**L 231: located instead of present.**

- We noted this correction.

**L 278 – 279: the 2001 image is from July, not exactly the end of the ablation season; could a possible overestimation of the IAs extent for that year be considered?**

- We agree that this can lead to a certain degree of overestimation, which will be hard to estimate since we do not have high-resolution images from this period from any other source for comparison. The thing that works in our favour; the images were acquired at the end of July, and looking at the amount of fresh snow remaining on the exposed rock faces, it is inevitable that the temperatures were relatively high this year. The weather records further corroborate this for 2001. This mitigates the risk of overestimation in mapping to a certain extent.

**L 312: a parenthesis is missed after the exponent of the second term (Ө).**

- We noted this correction.

**L 299-302: this introductive part could be shortened or omitted.**

- We shortened the introductory lines from this section as per your suggestion.

**L 345-346: this sentence is not clear, try to rephrase.**

- We noted this suggestion; we will rephrase this sentence in the revised manuscript.

**L 393-394: are 'erosion' and 'deposition' here referring to snow or to the avalanche activity? Please clarify.**

- Both the terms mentioned here refer to snow. We will make it clear in our text to avoid any further confusion.

**L 405: why is AdM weather record cited here? From section 3.3 one can understand that GSB record was used along with SAFRAN dataset, at least in this work. See comments for section 3.3 and try to clarify.**

- The same comment as for section 3.3.

**L 406: Sect. 3.3?**

- Correction noted.

**L 418: please, clarify what does it mean "annual temperature cycle".**

- The term 'annual temperature cycle' in our case refers to monthly mean temperatures estimated in $^{\circ}$C for an entire year. We will clarify this further in the text corresponding to this term to avoid ambiguity.

**L 449: equation (8)**

- Correction noted.

**L 465: check the consistency of the section numbers (e.g., DEM generation is described in sec. 3.1).**

- Thanks for pointing this out. We will check section numbers thoroughly for this particular line and the entire manuscript.

**L 469: …better to use "of the same surface" instead of "area".**

- Suggestion noted and incorporated.

**L 473-475: check for some repetitions here. (e.g., L 470).**

- The following text has been rewritten to avoid repetitions.

**L 485 (section 5.1): as with the comments in section 3.1, the text could also be summarized here, referring to what is reported in Kaushik et al. 2021, where the procedure is adequately described. In my opinion, the figures referring to this section (Fig. 6 and 7) should also be somewhat differentiated from those of Kaushik et al. 2021; perhaps the comment to the Fig. 6 might help (see below).**

**Alternatively, this part (section 5.1) could be summarized and included in a section of chapter 4 where an overall estimation of errors and uncertainties are provided, e.g. merged with the section 4.5.**

- Thanks a lot for this suggestion. We will summarize this section and cite our previous paper to shorten this section only detailing the essential information. We also agree with your suggestion that the images can be differentiated somewhat from our previous paper. We will also try to incorporate your suggestions for figure 6.

**L 535: add reference to Guillet and Ravanel (2020) instead of reference to the section 3.3.**

**L 538: located instead of present.**

- Suggestions noted.

**L 548: add a citation for this estimate.**

- We added Rabatel et al. 2013 as a citation for this estimate.
  Rabatel, A., Letréguilly, A., Dedieu, J.-P., and Eckert, N.: Changes in glacier equilibrium-line altitude in the western Alps from 1984 to 2010: evaluation by remote sensing and modeling of the morpho-topographic and climate controls, The Cryosphere, 7, 1455–1471, https://doi.org/10.5194/tc-7-1455-2013, 2013.

**L 550: exist at…**

- Suggestion noted.

**L 550: The reader remains a bit confused here. These numbers refer to the whole inventory of IAs (n=423), which is provided in Kaushik et al., 2021. However, only 200 IAs were considered in this work, i.e., a sub-sample of the complete inventory.**

**I would suggest being clearer, carefully distinguishing here and in the following part of the paragraph what refers to the previous work (with the full inventory) and what refers to the present work. Are the correlations similar when considering the full inventory and the sub-sample? Are there differences?**

- Dear reviewer, thanks for pointing out this critical issue. It is true that we have used a sub-sample inventory (n. 200) of the complete inventory (n. 423) for our comparison with meteorological and topographic parameters. The reason why we could not use the entire inventory stems from the fact that we were limited with data available for the years 1952 and 2001. We are limited to just one set of images for these time periods. Although the orthoimages are of considerably good quality and resolution, as with optical images, especially on steep slopes, the images suffer from shadow and illumination issues which can create uncertainty in our mapping estimates.
  Thus, after careful observation, we only selected the best IAs where we were certain about the precision of mapping, so we get a fair estimate of the correlation with meteorological and topographic parameters.
  However, as an important note, we also feel it is essential to report the uncertainty in our estimates for the total area of IAs, considering the complete inventory for the years 1952 and 2001. We are very sure of our mapping accuracy for 2012 and 2019 as we had alternate images for comparison. We will provide an uncertainty estimate (in +- % of the total area) for the total area of IAs estimated for 1952 and 2001.
  As for the correlation statistics, we checked the same for the complete inventory, but these correlations were comparatively lower for each parameter we considered, especially for comparison with the 1952 and 2001 images. The correlations are quite similar when we consider them between 2012 and 2019. This proved that the uncertainty was much higher for mapping with the older images.
  To clarify this point, we will include a paragraph detailing this difficulty in the revised manuscript to remove any confusion the reader would have.

**L 559: the reference is to Fig. 10d; in general, check the consistency between the text and the sequence of plots in Fig. 10.-**

- Thanks for pointing this out. We will thoroughly check this section's text and reference to the images.

**L 600-602: I do not find in section 4.2 what is stated here.**

- We added a statement in section 4.2 with reference from literature to justify the sentence here.

**L 604: Lopez et al., 2009 not reported in the reference list.**

- We will add this reference to the revised manuscript.

**L 609: not consistent with introduction (IAs area smaller than 0.1 km$^2$).**

- Thanks for pointing this out. We will correct this.

**L 610: Lopez et al., 2010 not reported in the reference list.**

**L 618: show the trends in Fig. 11.**

**L 665: in order to instead of better to?**

**L 680: IAs.**

**L 683: IAs.**

**L 680-682 and 683-685: rephrase these sentences avoiding repetitions.**

- Thanks for pointing out these corrections and your suggestion regarding L 680-682. We will incorporate these suggestions in the revised manuscript.

**Comments to figures and captions:**

**General comment to figures: check consistency between the figures regarding the font used for labels, numbers, and letters, etc.**

- Thanks a lot for this overall comment. We will check all images in detail and redraw them if necessary to have consistency between all the images.

**Fig. 1**

**I suggest using black squares to indicate the correct location of the two towns shown on the map. (i.e. Chamonix and Courmayeur).**

**The blue labels are not clearly visible, try changing the colour.**

**For scale and north arrow: leave some space between the frame and the graphic item.**

- We will redraw this image as per your suggestions.

**Fig. 3 and Fig. 4**

**Some labels should be enlarged to improve readability.**

- We will incorporate your suggestions regarding both images.

**Fig. 5**

**I suggest combining the four figures into a composite one, using the north arrow and legend only once.**

**The scale and coordinate labels should be enlarged.**

**Be consistent with Fig. 1 in the type of north arrow used.**

**The year of the orthophotos used as background should be reported.**

- Thanks a lot for your suggestions regarding this figure. We will incorporate all your suggestions regarding this image in the revised manuscript.

**Fig. 6**

**The same suggestion as Fig. 5: should be combined into a composite figure.**

**Add north arrow.**

- Suggestion noted. We will redraw this image.

**Fig. 10**

**The seven plots should be of the same size.**

**Some labels should be enlarged, and the size should be the same among the plots.**

**The legend could only be used once if the graphs were combined in a composite figure.**

- Thanks again for your suggestion. We will incorporate all the suggestions and build a new composite figure.

**Fig. 11**

**Check the label: Temperature (°C).**

**Some labels should be enlarged.**

- Suggestion and correction noted. We will change this.

**Fig. 12**

**The two plots could be combined in a composite figure, using the legend once, the same size of the frames and the same scale for the x-axis.**

**Figure caption: "Correlation between the mean normalized measured and modelled surface areas**

- Suggestion duly noted. We will try to combine both images in a composite image and modify the figure caption.

---

## Author Comment (AC2)

**General comments**

The paper is presenting the results of a study dealing with 70 years of evolution of ice aprons (IAs), namely cold ice fields located in very steep slopes (>40°), in the Mont-Blanc massive. 200 IAs have been investigated. As a main highlight, the paper evidences the dramatic and ongoing decrease in area of most IAs mainly in response to rising air temperature, with however an impact which is reduced and even not perceptible at the highest elevations.

The novelty of the paper is high as there has been almost no publication dedicated to IAs so far.

The paper is well structured and refers adequately to existing literature. Data, method and results are almost clearly presented. I would, however, suggest to separate the discussion aspects in a distinct section. The figures are mostly adequate, need however some improvement (see specific and technical comments). The conclusions are well concise and supported by the results of the study.

Besides an additional slight reorganization of the Results section (see Specific comments), my main concern is about the evolution of the accumulation proxy as a precipitation-temperature dependent factor and its impact on the IAs area changes. The proxy – which I fully agree with – is presented in the methodological section, but not further in the results (sub-section 5.5).

I would consider the paper very worthy of being published, after minor improvements

having been undertaken.

**REPLY:**

Dear reviewer, thanks a lot for taking the time to meticulously review our paper and providing valuable feedback to improve the manuscript. Your encouraging response motivates us to improve the manuscript further, considering all your suggestions and feedback. As per your suggestion in this general comment regarding the separation of the results and discussion section, we completely agree with this. We assure you that the revised manuscript we submit will consider this specific comment from you. We will further work towards improving the quality of our images as per your suggestions. You will find our replies to the specific and technical comments you have on the manuscript below. Thanks in advance also for taking the time to read the revised version of the manuscript.

**Specific comments:**

**As ice aprons are almost unknown in the literature and to facilitate the understanding, I would strongly suggest to insert an initial figure (picture) illustrating what is talking about. For sure, many very illustrative pictures should exist. The orthoimages presented in Fig. 5 are not sufficient for that purpose.**

- Thanks a lot for this specific comment. We agree that an initial high-resolution figure would be helpful for a better understanding of ice aprons. We have many images and field photographs of ice aprons which we can insert in the initial sections of the manuscript. We will add some high-resolution photographs for this purpose in the revised manuscript.

**There is only one year used for the longer-term analysis, namely 1952. The conditions during that year could be worth of being described. According to the GSB data, there was a severe heat wave of a few weeks in late June – early July. 1952 was also finishing a period of about 10 warmer years with some "hot" summers as 1947. A significant reduction of IAs took place during those years, before that the conditions became again more favorable for the next about three decades. This is attested for some alpine IAs outside of the MBM area (e.g. Mont-Blanc de Cheilon in the Valais Alps).**

- This is a critical comment, and we fully agree with your suggestion. it is important to discuss the weather during this particular year to have a better idea of the evolution of ice aprons during this year. We will add a paragraph in section 3.3 to discuss this specifically in the revised manuscript.

**(L. 425) I agree with the way of doing for estimating the accumulation on ice aprons being limited to precipitation by air temperature ranging between -5 and 0°C. It would be nice to provide an example for an annual period, at different elevation. It will show that such conditions are only (mostly) prevailing during the summer half-year and the winter precipitation are almost not entering into consideration (what is maybe however not the case on south slopes). Later in the result section, similarly to figure 11, it would be important to illustrate its evolution since 1952.**

- Dear reviewer, thanks for raising this issue. We agree with your comment regarding the conditions being favourable for precipitation mostly in the summer months. We have added a

new plot for the year 2019 to show this as per your suggestion. The plot (also attached at the end of this document) clearly shows precipitation values are highest in the summer months. Some precipitation at lower elevations occurs in winter, but at higher elevations, most precipitation occurs only in the summer. This new plot can be added to the revised manuscript as per your suggestion.

Coming to the other suggestion for the accumulation proxy and its evolution since 1952, we initially thought of having this plot as part of our manuscript, but in contrast to the PDD plot (figure 11), this plot is haphazard and does not provide any clear information. We attach this figure also at the end of the manuscript for your reference. Instead, working on your other suggestion, we can provide a figure showing the rate of accumulation at all elevations between -5 and 0 °C for each time period (1952 to 2001, 2001 to 2011 and 2011 to 2019). As can be seen from the plot, accumulation rates are decreasing in general over the years. This can be a piece of relevant information, in our opinion. We attach the new figure also at the end of the manuscript for your reference.

**A description of the spread of IAs over e.g. elevation and aspect is missing. There is the**

**figure 1, but it does not help.**

- We agree that this information is useful. For this inventory of 200 IAs, 77% of the IAs are located above the 3200 m a.s.l. (the regional ELA, according to Rabatel e.al, 2013), while a majority of them exist between 3200 and 3800 m a.s.l. (63 % of the total count). For the aspect, a majority of the IAs are located in the northern aspects i.e. N, NE and NW (55 % of the total), while the eastern aspects are the least dominant. We will add this to the revised manuscript along with a couple of graphs (attached at the end of this document) A big discussion on this topic is also a part of the paper which we have under review currently, while this information is also published for the entire database in Kaushik et.al 2021.

**In the result section, beside a sub-section 5.3 is missing, I would suggest to invert the**

**order of sub-sections 5.5 (Influence of changing climate…) and 5.4 (Influence of local**

**topography…), as 5.5 appears to be more closely the follow-up of sub-section 5.2 (Total**

**loss area… over seven decades).**

- Thanks for pointing out the mistake with missing section 5.3. We also agree with the suggestion that section 5.5 can come before section 5.4 in the manuscript. We will correct this in the revised manuscript.

**Minor/technical comments:**

**L.83 – Maybe replace "most" by "many" or "frequent", or does it apply to the MBM area only ?**

- Suggestion noted and incorporated in the revised manuscript.

**L.227 – What is the source of the GSB data? Is the homogenized time series used? Because there is quite a significant difference from the non-homogenized data.**

- Dear reviewer, the GSB data comes from MeteoSwiss (https://www.meteoswiss.admin.ch/home/climate/swiss-climate-in-detail/homogeneous-data-series-since-1864.html?station=gsb).

Yes, the data we have used is homogenized, available from 1864 till today. We will also mention this important point in the manuscript.

**L.243 – Using the GSB data for precipitation as a proxy for the MBM area is a bit tricky, as the GSB pass is located in the "shadow" of the MBM massive by "westerlies" and is largely influenced by precipitation coming from the south. But I know, this is difficult to do better.**

- We agree there is a bit of uncertainty generated because we use the GSB data to calculate our precipitation proxy (especially from 1952 to 1959), where we lack the SAFRAN datasets. Although Col du Saint Bernard in Aosta valley represents almost similar conditions to the MBM, as you mentioned, certain site-specific differences exist which are hard to overcome and somewhat beyond our control. We are unfortunately also limited with data available from any other source for such a long term.

**L.330 – Figure 3 is not depicting precipitation.**

- Thanks for pointing this out. We have revised this in the manuscript and mentioned temperature here.

**L.510 – There is some issue with the values of area loss and their correspondence to**

**figure 8 :**

**The area reduction is 2001 compared to 1952 is 25.4 % and must be rounded to -25%**

**31% is the relative area loss in 2019 compared to the 2001 area (this must be specified)**

**or recalculated to the 1952 area. In addition, this value is wrong as it is obviously the**

**addition of the area reduction in 2012 to 2001 and in 2019 to 2012. The area reduction in**

**2019 compared to 2001 is 28.8%**

**The "alarming rate" is not provided but left to the calculation by the reader. The values**

**must be presented, for instance as an average annual rate compared to 1952, which**

**appears to be 0.5%/year from 1952 to 2001, 1.1%/year from 2001 to 2012 and**

**1.2%/year from 2012 to 2019.**

- Thanks a lot for pointing out this critical error in the text. We corrected this in our text in section 5.2. We have also mentioned the area loss rate, as you suggested in the revised manuscript.

**L.617-618 - The evolution of the accumulation rate must be provided as well**

- Dear reviewer, as per the previous comment, we can add a new graph in the manuscript to show the evolution of the accumulation rates and mention this also in the text.

**L.636 – 4 IAs are increasing is size. Is this significant for all ? Where are these 4 IAs**

**located ? Maybe worth of providing a picture of each ?**

- Dear reviewer, thanks for pointing this out. It will be relevant to give information about these IAs. This is not significant as other IAs do not show the same trend. But nevertheless, for future analysis these four IAs could be of interest. The 4 IAs in the question here are: 2 IAs on the N and NW face of Rochers Rouges Inferieurs (~4350 m a.s.l. and 4050 m a.s.l.) near the Grand Plateau, 1 IA on the NE face of Col de la Brenva (~4160 m a.s.l.) and 1 IA on the S face of Col du Bionnassay (~4050 m a.s.l.). As observed, all these IAs are located at elevations higher than 4000 m a.s.l. It can be expected that a few IAs could show an increase in surface. However this increase in surface area is not dramatic (~10 % increase in surface area).
  The pictures of all these IAs can be provided, however since the number of images in our manuscript is already very high, maybe we can provide them as supplementary material instead of adding them in the manuscript.

**L.678 – The comment on L. 510 must be considered and the sentence adapted in**

**Accordance**

**L.680 – The "climate forcing parameters" must be specified**

**L.683 – 685 – This bullet can be omitted at it would not say anything else than the next**

**one, but keeping vague ("some topographic factors… , while other factors…")**

- All the following comments were noted and incorporated into the revised manuscript.

**Figures**

**As a general comment for the figures: the layout must be improved for many of them.**

**The legibility must be checked, the character size must be homogenized and made large**

**enough, the use of caption and brackets in the axis legends must be homogenized, all**

**unnecessary surrounding boxes (e.g. fig. 9 – 10) should be removed.**

- Dear reviewer, thanks for this general remark. We will consider all your suggestions and try to homogenize all the images in the revised manuscript.

**Figure 3 – The figure is not legible. If it is meant to show an annual cycle at different**

**elevations, it must provide just one year (which could be the mean 1952-2019). If it is**

**meant to show the overall trend, only a running annual (or multiannual) mean should be**

**represented.**

**Why 8/1/1952 in the time axis ?**

**What does the box "Interpolated data from GSB temperature" mean ? Better to insert an**

**arrow to the 1952-1958 box.**

**There are two issues with the blue (2400 m) curve. First, it is mostly shifted in comparison**

**to the other (e.g. for the last years, the peak temperature is appearing in winter). Second,**

**there is a peak temperature apparently in 1985, which has never occurred. There is a**

**mistake somewhere. July 1983 was extremely hot, but nothing occurred in 1985.**

- Dear reviewer, thank you for pointing out this error with the graph. We agree with your suggestion completely as the previous graph did not provide much information. We have

made a new figure to show the multiannual variation of mean temperatures from 1952 to 2019. The new graph shows mean annual temperature values for every year from 1952 to 2019. The figure is attached at the end of the document for your reference.

**Figure 4 – Again, there is an outlier at about +14°C (in SAFRAN), which is doubtful. This is probably the 1985 peak mentioned above… but why not at the same temperature (+15°C in Fig. 3) ?**

- We cross checked this value again from the SAFRAN data. The value (13.7 $^{o}$C) actually comes from August 2020. We also believe this is an outlier value not truly representative of the actual temperature during this month. But, SAFRAN data shows this value. We can remove this data point as 2020 is not part of the observed study period and give a new figure in the revised manuscript.

**Figure 5 - Yellow on white is not adequate. Maybe orange ?**

- We will redraw these images again, taking into consideration your comment.

**Figure 10 – The layout (legend, axis label, dot size, etc) must be homogenized and made legible on all figures.**

- We will redraw all these images taking into consideration your comments.

**Figure 11 – What are the represented values ? What is for instance a PDD ranging from +14 to +30°C at 2400 m ? I don't understand.**

- Dear reviewer, the values represent annual PDD values in $^{o}$C. This is to show the evolution of the temperature proxy since 1952. We realized the axis label on the y-axis should be PDD ($^{O}$C) to avoid confusion. We changed this in the revised manuscript. PDD ranging from 14 to 30$^{o}$C is the annual PDD value for each year (the sum of all positive mean monthly temperatures in one year).

**Figure 12 – Legend … "The colour and size of the ticks represent the mean elevation of the IA". I guess the colour one is representing the elevation, the dot size being representative**

**of the IA size (in this case, the legend must be provided)**

- Dear reviewer, in our case, both the size and colour represent the mean elevation of the IA. We realize it is probably unnecessary to represent the same information in two different ways. It is just for better visual interpretation. We can change all ticks to the same size if it creates unnecessary confusion.

[Figure]

(a)                                                                  (b)

Figure: Distribution of the IAs with: a.) elevation and b.) Aspect

[Figure]

Figure: Variation of mean annual temperatures from 1952 to 2019 at different elevations.

[Figure]

Figure: Variation of precipitation occurring between -5 and 0°C at different elevations from 1952 to 2019.

[Figure]

Figure: Variation in the total precipitation occurring between -5 and 0°C through 2019 at different elevations.

[Figure]

Figure: Variation in the accumulation rates at different elevations for each time period of observation

---

## Referee Report (RR1)

[referee-annotated manuscript omitted]